# Mapping of Determinants of Urinary Sex Steroid Metabolites During Late Pregnancy: Results from Two Spanish Cohorts

**DOI:** 10.3390/ijms262311598

**Published:** 2025-11-29

**Authors:** Emily P. Laveriano-Santos, Estelle Renard-Dausset, Mariona Bustamante, Dolors Pelegri, Zoraida García-Ruiz, Marina Ruiz-Rivera, Marta Cosin-Tomas, Elisa Llurba-Olive, Maria Dolores Gomez-Roig, Noemi Haro, Óscar J. Pozo, Payam Dadvand, Martine Vrijheid, Léa Maitre

**Affiliations:** 1ISGlobal, 08003 Barcelona, Spain; emily.laveriano@isglobal.org (E.P.L.-S.); estelle.renard@isglobal.org (E.R.-D.); mariona.bustamante@isglobal.org (M.B.); dolors.pelegri@isglobal.org (D.P.); zoraidagarciaruiz@gmail.com (Z.G.-R.); marina.ruiz@isglobal.org (M.R.-R.); marta.cosin@isglobal.org (M.C.-T.); payam.dadvand@isglobal.org (P.D.); martine.vrijheid@isglobal.org (M.V.); 2Centro de Investigación Biomédica en Red Fisiopatología de la Obesidad y la Nutrición (CIBEROBN), Institute of Health Carlos III, 28029 Madrid, Spain; 3Universitat Pompeu Fabra, 08002 Barcelona, Spain; 4Spanish Consortium for Research and Public Health (CIBERESP), Instituto de Salud Carlos III, 28029 Madrid, Spain; 5Institut de Investigació Biomèdica Sant Pau—IIB San Pau, 08041 Barcelona, Spain; elisa.llurba@uab.cat; 6Clinical and Epidemiological Neuroscience (NeuroÈpia), Institut d’Investigació Sanitària Pere Virgili (IISPV), 43204 Reus, Spain; 7Hospital del Mar Research Institute, 08003 Barcelona, Spain; nharo@researchmar.net (N.H.); opozo@researchmar.net (Ó.J.P.); 8BCNatal—Barcelona Center for Maternal, Fetal and Neonatal Medicine, Hospital Sant Joan de Déu and Hospital Clínic, University of Barcelona, 08028 Barcelona, Spain; lola.gomezroig@sjd.es; 9Institut de Recerca Sant Joan de Déu, 08025 Barcelona, Spain

**Keywords:** steroidomics, maternal metabolome, metabolic yprofiling, urinary steroid prolife, phase II metabolism, conjugated metabolites

## Abstract

Steroid hormones (SHs), including sex steroids and corticosteroids, are crucial for a healthy pregnancy. We aimed to comprehensively characterize the maternal SH metabolome in late pregnancy and identify clinical, lifestyle, and sociodemographic determinants influencing SH metabolism with a replication in an independent cohort. Urinary SH metabolites were analyzed in 1221 third-trimester pregnant women (aged 28 to 37 years) from two Spanish cohorts, BiSC (2018–2021, *n* = 721) and INMA-Sabadell (2004–2006, *n* = 500), using targeted UHPLC-MS/MS. We quantified 50 SH metabolites, resulting in 13 hormone groups, 9 sulfate/glucuronide ratios, and 17 estimated steroid enzymatic activities across steroidogenesis pathways. We applied elastic net regression to identify determinants, and multivariable linear regression models to estimate variance explained. Among the 47 and 28 determinants from BiSC and INMA-Sabadell, respectively, 10 determinant-SH metabolome pairs showed statistically significant associations (*p* < 0.05), supporting robust replication. Maternal BMI was the main determinant linked to higher corticosteroid and androgen metabolites. Higher physical activity was associated with lower glucocorticoids and progestogen metabolites, while older maternal age was related with lower levels of androgen and corticosteroid metabolites. Tobacco exposure in the first trimester predicted higher levels of cortisol metabolites. Latin American women had lower cortolone levels compared with Spanish women. Parity, dietary fat intake, sleep, alcohol intake, and sex of the fetus contributed to smaller variations in different SHs. This dual-cohort analysis provides the most detailed and replicated evidence to date of how clinical, lifestyle, and sociodemographic factors shape the maternal SH metabolome during late pregnancy.

## 1. Introduction

Steroid hormones (SHs) play a crucial role throughout pregnancy, regulating metabolic, immune, and stress-related processes, as well as supporting fetal growth and development [1]. These hormones include sex steroids such as androgens, progestogens, and estrogens, as well as corticosteroids, including glucocorticoids and mineralocorticoids (Figure 1) [1,2,3,4,5,6]. During gestation, the placenta acts as the primary endocrine organ, producing large amounts of estrogens and progestogens, while the fetal adrenal cortex generates androgen precursors that are further converted into estrogens [1]. SH levels increase during pregnancy, especially in the third trimester, when estrogens and progestogens rise to sustain fetal development [1,2].

Determinants that explain the interindividual variability in SH levels during pregnancy have been described in the literature. Maternal age, body composition, genetic variants, lifestyle factors such as diet, smoking, physical activity, and stress, and fetal sex are the determinants reported in several studies [7,8,9,10,11,12,13,14,15,16,17,18,19,20,21,22,23,24,25,26,27,28,29,30]. In a large study of 949 non-pregnant adults, Deltourbe et al. (2025) reported that sex, age, body mass index, smoking, hormonal contraceptive use, and genetic variants strongly affect circulating SH concentrations [31]. However, most studies to date have been limited to serum measurements of a few selected hormones and have not captured upstream intermediates or conjugated metabolites that could reveal broader patterns of SH metabolism.

A comprehensive characterization of SH during pregnancy requires evaluating phase I and phase II metabolism, including conjugated metabolites such as glucuronides, monosulfates, and disulfates [32]. While plasma and serum are the most common matrices for SH quantification [33], urine represents a practical, noninvasive, and repeatable alternative for epidemiological studies [34,35]. Urinary SHs reflect the end products of renal and hepatic SH metabolism and thus provide integrated insights into enzyme activity, hormone clearance, and endocrine regulation [32,35]. Nevertheless, large differences have been reported among studies, with SH concentrations varying by up to 5000-fold within the same biological matrix and gestational period, reflecting inconsistencies in analytical methods [33,36]. This heterogeneity complicates direct comparison between studies and replication, particularly in metabolomics research, where analytical reproducibility is essential [37,38].

Although determinants of SH variability have been described and targeted analytical methods are available, there are no studies on later pregnancy that have quantified a broad panel of conjugated SH metabolites and replicated their findings in an independent cohort using harmonized analytical approaches. To address this gap, we quantified 50 urinary SH metabolites spanning all major steroidogenesis pathways (Figure 1, Appendix A) using a targeted LC-MS/MS approach in pregnant women in their third trimester of gestation from the richly phenotyped Barcelona Life Study Cohort (BiSC). Determinants of SH were explored using an Exposome-Wide Association Study (ExWAS) and elastic net (ENET) regression to identify hormone-specific predictors, including clinical, genetic, sociodemographic, lifestyle, and fetal factors. For each SH, linear models were then fitted using only the predictors selected by ENET to estimate their relative contributions. To confirm the robustness of our findings, we replicated our analysis in an independent cohort, INfancia y Medio Ambiente (INMA)-Sabadell, using identical laboratory procedures, data processing pipelines, and analytical methods. Together, these data provide a comprehensive characterization of maternal SH metabolism in late pregnancy and its key determinants.

## 2. Results

### 2.1. Participants’ Characteristics

A total of 721 participants from the BiSC study were included in the primary analysis, while 500 participants from the INMA-Sabadell study were included in the replication analysis (Figure 2, Appendix A).

The mean maternal age in BiSC was 34.4 years (SD 4.4), and urine samples were collected at a mean gestational age of 34.7 weeks (SD 1.5). Most participants had normal weight (64%), were Spanish (69%), had a university education (72%), and were nulliparous (61%) (Table 1). For the replication analysis, 500 participants from the INMA-Sabadell cohort were analyzed. The mean maternal age was 31.2 years (SD 4.3), and the mean gestational age at urine collection was 34.3 weeks (SD 1.6), where 66% were of normal weight, 89% were Spanish, 73% had higher education, and 57% were nulliparous (Table 1).

In both cohorts, over 85% of participants reported no alcohol consumption or smoking during pregnancy or at its onset. Most participants reported sleeping more than 7 h per day (79% in BiSC and 87% in INMA-Sabadell), and taking folic acid supplements during pregnancy (74% and 82%, respectively). The mean Mediterranean Diet score was similar in both cohorts (3.9 ± 1.8 in BiSC and 4.3 ± 1.5 in INMA-Sabadell). Regarding contraceptive use prior to the last pregnancy, 87% of BiSC participants and 90% of INMA-Sabadell participants reported using any method. Hormonal contraception was reported by 20% of BiSC and 35% of INMA-Sabadell participants (Table 1). Correlation patterns among determinants are shown in the Appendix A.

### 2.2. Identification of SH Metabolites

Of the 57 SHs (sex steroids and corticosteroid metabolites) detected in urine, 7 were excluded from the analysis due to their poor reproducibility (Relative Standard Deviation > 35%). These included 5α-androstan-3β 17β-diol-diSulfate -I, -II, and -III, pregnandiol-sulfoglucoconjugate, 5-pregnenolone-sulfate, 5-pregnendiol-sulfate, and cholesterol sulfate. A total of 50 SH metabolites were considered in the current analyses (Appendix A), including the androgen, estrogen, progestogen, corticosteroid families from the steroidogenesis pathways (Figure 1).

The raw levels of the 50 SH metabolites are described in Appendix A and Table 2. These metabolites were also aggregated into 13 indicators of the total production of SH families or subfamilies (Appendix A), 9 sulfate/glucuronide (S/G) ratio (Appendix A), and 17 indicators of steroidogenic enzymatic activity calculated as the ratio between product and precursor (Appendix A), with a total of 89 SH molecular features considered as outcomes. For the statistical analysis, all SH levels were adjusted for gestational week when urine sample was collected using the multiples of normal median (MoM) approach. Progesterone and pregnenolone were the predominant groups of metabolites in both cohorts, accounting for over 80% of total SH concentrations (Figure 3).

In addition, pairwise correlation analysis showed positive correlations between the steroid metabolites from each family in the BiSC (Appendix A). The androgens total dehydroepiandrosterone (DHEA) and total testosterone had a positive correlation with each other (r = 0.26, *p* < 0.0001) and with total progestogens (r = 0.24, *p* < 0.0001). Given that testosterone is a precursor for estrogens that can be converted into each other, their positive correlations were also expected (r = 0.19, *p* < 0.0001).

### 2.3. Main Determinant Selection of the SH Metabolome (BiSC): ExWAS and ENET Analyses

We first systematically tested the association between each determinant variable and each SH molecular feature through an ExWAS (Appendix A). Overall, we tested more than 4800 associations between exposure and SH molecular features (54 exposures (including dummy variables), *89 outcomes (50 SH metabolites, 13 indicators of the total production of steroid families or subfamilies, 9 S/G ratio, and 17 indirect indicators of steroidogenic enzymatic activity calculated as the ratio between product and precursor)). ExWAS linear regression models were adjusted by technical variables related to urine collection such as hospital visit in the third trimester or at birth, COVID-19 exposure period (COVID confinement), and season of birth. After the ExWAS analysis, 43 determinants were significantly associated with at least 1 of the 89 SH molecular features, yielding 236 positive and 265 negative suggestive associations (*p* < 0.05, Appendix A and Figure 4). Further multiple testing correction using the effective number of tests (ENT) was applied for each determinant within each SH molecular features (see Methods for more details). With this criteria, 47 determinant-SH molecular feature associations were statistically significant comprising 17 positive and 30 negative associations (Appendix A). Miami plots display determinant-SH molecular feature associations by family of determinant (Appendix A). Maternal body mass index (BMI) in the third trimester was the main determinant associated with SH molecular features (60% of the associations observed after multitesting correction), followed by maternal age (17%), energy-adjusted fat intake (9%), ethnicity (4%), fetal weight in the 3rd trimester (4%), adjusted Mediterranean diet (aMED) (2%), Polycystic Ovary Syndrome (PCOS) (2%), and progesterone medication (2%) (Appendix A).

To account for potential collinearity between predictors and to identify the most influential variables in a multivariable context, we applied ENET, a penalized regression method that combines L1 and L2 regularization to perform variable selection and shrinkage simultaneously (see Methods for more details). Using ENET, we identified 45 determinants associated with at least 1 of 86 SH molecular features, including 49 individual SH metabolites, 11 summed families or subfamilies, 9 S/G ratios, and 17 phase I enzymatic activities (Figure 5, Appendix A). Three SH molecular features (5-Pregnendiol-DiSulfate (5PD-diS), total pregnenolone, and total progestogens) were not associated with any determinant (Figure 5).

### 2.4. Determinants of Steroid Hormones: Relative Importance Analysis in the BiSC

Out of the 45 determinants tested, we identified significant associations with at least 1 out of the 86 SH molecular features for 37 predictors, in the categories of clinical parameters, lifestyle, mental health, sociodemographic, genetic polymorphism, and fetus predictors (Appendix A). The predictors contributed to up to 17% of the variance explained from the models (Figure 6).

We observed that among all the clinical determinants, maternal BMI in the third trimester made an important contribution to multiple SH molecular features. Higher BMI was significantly associated with 46 different SH molecular features, including higher total corticosteroids and lower total estrogens and individual progestogen metabolites. Regarding S/G ratio, we observed that BMI was associated with a lower S/G ratio of corticosteroids, progestogens, and androgens, indicating a lower concentration of these SH metabolites in their inactive form. Finally, BMI was also positively associated with several types of phase I enzymatic activity implicated in progestogen, androgen, and corticosteroid metabolism. Gestational weight gain also contributed substantially, being associated with 24 SH molecular features. Most of these associations were negative and involved androgens, estrogens, and progestogens, suggesting that higher weight gain is related to lower levels of these SH metabolites. An exception was observed for epiandrosterone-sulfate (epiAN-S), which showed a positive association with weight gain. Some determinants related to obstetric complications (OC) were associated with some SH molecular features. Mild preeclampsia was linked to higher levels of androgen and corticosteroid metabolites, as well as lower S/G ratios for progestogens and progesterone. In contrast, type I diabetes was associated with consistently lower levels of corticosteroids. Pregnant women with a PCOS diagnosis before pregnancy had higher androgen levels. Parity was also an important determinant of SH molecular features. Compared with nulliparous women, primiparous women had lower levels of several corticosteroid metabolites, lower estrogen (E1-G), and reduced 5α-reductase activity. In addition, medication use had an influence on SH metabolism: corticosteroid treatment was associated with lower estrogen levels, whereas progesterone treatment was linked to lower levels of corticosteroid metabolites but higher levels of testosterone metabolites (Appendix A).

Among lifestyle determinants, higher energy-adjusted fat intake in the second trimester was one of the main dietary predictors associated with higher levels of androgens, progestogens, and S/G ratios of estrogens and progestogens. Higher adherence to the Mediterranean diet (aMED) was associated with increased S/G ratios of corticosteroids, cortisol, and androgens. In contrast, moderate alcohol intake (0.4–1 g/day) during the second trimester was linked to lower progestogen metabolites and 20α-reductase activity, while one corticosteroid (F-S) was higher compared to abstainers. Maternal smoking was also an important predictor: women who smoked more than 10 cigarettes per day in the first trimester had higher levels of androgen, progestogen, and corticosteroid metabolites compared to non-smokers. Finally, moderate-to-vigorous physical activity (MVPA) exceeding 150 min/week in the third trimester was consistently associated with lower levels of corticosteroid metabolites (Appendix A).

Among sociodemographic determinants, increasing maternal age was associated with lower levels of androgens, estrogens, and progestogens. Ethnicity was also a strong determinant of SH molecular features: compared with Spanish women, those born in Latin America had lower progestogen and corticosteroid levels, and higher S/G ratios for corticosteroids and androgens. Regarding maternal education, women who had completed university-level studies had lower progesterone metabolite levels but higher total corticosteroid levels compared with women with a primary education (Appendix A).

Regarding genetic variants, some maternal single-nucleotide polymorphisms (SNPs) were identified as significant determinants of SH molecular features. The rs2300701 at the *SRD5A2* gene was associated with higher total androgens and total testosterone. rs1937863 at the *AKR1C2* gene was linked to higher estradiol sulfate (E2-S) levels. rs6493497 at the *CYP19A1* gene was associated with higher progestogen metabolites and increased S/G ratio of progesterone. rs743572 at the *CYP17A1* gene showed associations with individual androgens and corticosteroids, whereas rs806645 at the *SRD5A2* gene was associated with higher corticosteroid metabolites. Finally, rs700518 at the *CYP19A1* gene was associated with lower 11β-hydroxysteroid dehydrogenase type 2 (11β-HSD2) activity (Appendix A).

Among maternal mental health indicators, high depressive symptoms in the third trimester were associated with higher androgen levels (16b-hydroxy-DHEA-diSulfate (16OH-DHEA-diS) 2, 5-androsten-3b17b-diol-diSulfate(5AD-diS) -1, testosterone) and slightly lower 3β-hydroxysteroid dehydrogenase (3βHSD) activity, while higher stress score showed higher levels of androgen and corticosteroid metabolites (Appendix A).

Finally, among fetal-related determinants, fetal weight at 20 weeks was significantly associated with multiple SH metabolites. Higher fetal weight predicted higher levels of corticosteroids and estrogen metabolites. In contrast, fetal weight was inversely related to androgens, including DHEA sulfate and 5-androsten-3b17b-diol-diSulfate, and was negatively associated with global CYP17 activity, suggesting suppressed androgenic enzymatic activity. Fetal sex was also a significant determinant of steroid metabolism. Pregnancies with female fetuses were characterized by higher levels of estriol metabolites including estriol-glucuronide (E3-G), and higher activity of CYP17A1 and 3β-HSD enzymes compared to male fetuses. Pregnancies with female fetuses also had lower levels of 17-hydroxy-5-pregnenolone-3-sulfate, a progestogen metabolite (Appendix A).

### 2.5. External Replication Analysis (INMA-Sabadell)

We replicated the association between determinants and SH molecular features in the INMA-Sabadell cohort, which included 500 participants with 34 weeks of gestation at urine sampling collection (Appendix A). Using the determinants selected in the primary analysis and those available in the INMA-Sabadell cohort, 28 of 47 determinants and 86 SH molecular features were considered in the regression analysis. A total of 10 determinants were significantly associated with at least one SH molecular feature, and in most of the cases, the direction of associations was consistent across cohorts, supporting the robustness of our findings (Appendix A, Figure 7 and Appendix A).

Regarding clinical determinants, maternal BMI (before pregnancy) was the main clinical determinant of SH metabolism. Higher maternal BMI was associated with higher corticosteroid metabolites. At the enzymatic level, maternal BMI was positively associated with 21-hydroxylase, 20β-reductase, 5β-reductase + 3α-HSD, CYP17, 17,20 lyase, and 11β-HSD2, whereas 11β-HSD1 and aromatase showed inverse associations. Maternal BMI was also linked to lower S/G ratios for total corticosteroids, cortisol, and testosterone. Regarding parity, primiparity was linked to lower corticosterone sulfate but with higher 20β-Cortolone-glucuronide (cortolone-G2) levels (Appendix A and Figure 7).

Among lifestyle determinants, higher physical activity during the first trimester was associated with lower total corticosteroids and total cortisol, pregnantriol-glucuronide (PT-G) and total 17-hydroxyprogesterone levels. Sleeping more than 7 h per day during the first trimester was linked to lower SG ratios of testosterone. Tobacco exposure in the first trimester predicted higher levels of cortisol metabolites. Moderate alcohol consumption (0.4–1g/day) in the first trimester was linked to higher levels of 5α-Pregnandiol-3β-sulfate-20α-glucuronide (PD-SG_1) compared to non-consumers. Energy-adjusted dietary fat intake in the third trimester was inversely associated with 5-androsten-3α17β-diol-disulfate (5AD-diS_2) and 16α-hydroxy-DHEA-sulfate (16OH-DHEAS_2) (Appendix A and Figure 7).

Among sociodemographic determinants, maternal age and ethnicity were significant predictors of SH molecular features. Higher maternal age was associated with lower levels of androgen metabolites including 5AD-diS_2, epiAN-S, androsterone-glucuronide (Andros-G), androsterone-sulfate (AN-S), 16OHDHEA-diS_1, etiocholanolone-glucuronide (Etio-G), total DHEA, and total androgens. Higher maternal age was also associated with lower levels of cortisone-sulfate (E-S), pregnantriol-diSulfate (PT-diS), and 21-Hydroxypregnenolone-diSulfate (21OH-5P-diS). Regarding ethnicity, women of Latin American origin had lower Etio-G, cortolone-G2, and 20β-reductase activity and higher S/G ratios for corticosteroids compared with Spanish women (Appendix A and Figure 7).

Fetal determinants also contributed to variation in maternal SH metabolism. Female sex was associated with higher 3β-HSD activity and lower 17-hydroxy-5-pregnenolone-3-sulfate (5PD-20one-S) levels (Appendix A and Figure 7).

## 3. Discussion

This study provides one of the most comprehensive characterizations of the maternal SH metabolome during late pregnancy, integrating data from two large and well-characterized Spanish cohorts. Our analysis identified that maternal clinical, lifestyle, sociodemographic, genetic, and fetal factors contribute to interindividual variability in SH metabolism.

### 3.1. A New Window into Maternal Steroid Metabolism

Using a highly sensitive targeted metabolomic LC-MS/MS approach, we identified 50 conjugated urinary SH metabolites across major SH classes, including glucocorticoids, androgens, estrogens, and progestogens. Beyond measuring individual metabolites and total class levels, we also estimated functional indices of SH metabolism, such as phase I enzymatic activities and S/G ratios. Previous studies have largely focused on free or unconjugated SH fractions in plasma or serum, using immunoassays, LC-MS/MS, or GC-MS [33,39,40]. However, few studies have considered conjugated metabolites, including glucuronides, sulfate, or sulfoglucoconjugate forms, which represent major end products of SH metabolism [32,41]. Our study extends this research by reporting novel metabolites such as pregnanediol sulfoglucoconjugate isomers (5-pregnendiol-SG) and estriol-sulfoglucoconjugate, as well as some bisulfate SH metabolites, a minor fraction of the urinary SH metabolome.

The urinary SH profile described in our study aligns with previous reports of maternal SH trajectories during pregnancy. In a recent study of 127 Spanish pregnant women, Servin-Barthet et al. (2025) quantified 34 urinary glucuronide and sulfate SH metabolites across multiple time points from pre-pregnancy to postpartum [41]. Consistent with our findings, estriol glucuronide (E3-G) was the predominant estrogen metabolite in the third trimester. For corticosteroids, we identified 20α-cortolone-glucuronide (cortolone-G1), cortolone-G2, and tetrahydrocortisone-glucuronide (THE-G) as major metabolites, whereas Servin-Barthet et al. (2025) reported THE-G and 11-dehydrocorticosterone sulfate as predominant metabolites [41]. Similarly, Andros-G and Etio-G were the main androgen metabolites across both studies. Progressive increases in urinary corticosteroid, progestogen, and estrogen glucuronides during pregnancy since 26 weeks of gestation until before delivery was described by Jänti et al. (2013) [42]. These observations are in agreement with longitudinal plasma and serum studies reporting gradual increases in estrogens, progestogens, and corticosteroids during late gestation [33,43]. The physiological increase in SH levels during pregnancy, followed by a decrease postpartum, represents a well-established endocrine adaptation essential for maternal and offspring health [44].

### 3.2. Clinical Factors Are Consistently Associated with SH-Level Differences

Maternal BMI was the main determinant of SH metabolism in both cohorts. Higher BMI was associated with higher urinary androgens and glucocorticoids, alongside higher activity of 21-hydroxylase, 20β-reductase, and 11β-HSD2, enzymes that regulate corticosteroid and androgen metabolism [36,45]. Higher maternal BMI was also associated with lower aromatase activity, a critical enzyme that mediates the conversion of androgens to estrogens [36]. In a multi-center study conducted on a cohort of 548 US pregnant women carrying singletons, higher maternal BMI was associated with lower serum estrogen and higher testosterone in the first trimester (median 12 weeks) [14]. In a systematic review based on fifteen studies, Volqvartz et al. (2023) concluded that high maternal BMI downregulated the placental 11β-HSD2 activity, increasing the fetal exposure to active cortisol [46].

Parity was another clinical determinant associated with corticosteroid metabolism, but the pattern differed between cohorts. In the BiSC, primiparous women had lower corticosterone and cortolone-G1 levels compared with nulliparous women. In the INMA-Sabadell cohort, primiparity was linked to higher cortolone-G2 (cortolone-glucuronide isomer) levels. Cortisol levels during mid- and late pregnancy tend to be higher in primiparas compared with multiparas [47,48] and were partially mediated by pregnancy distress [47].

### 3.3. Lifestyle Behaviors: Physical Activity, Sleep, Smoking, Alcohol, and Diet Are Determinants of SH Levels

Physical activity was the lifestyle determinant with the largest contribution to maternal SH metabolism. In both cohorts, higher MVPA or total activity (METs) was associated with lower cortisol and corticosteroid levels. These findings suggest that physical activity during pregnancy can regulate maternal corticosteroid metabolism. Budnik-Przybylska et al. (2020) observed a negative correlation between hair cortisol and exercise frequency in 29 pregnant women [49]. Rauramo et al. (1982) reported that progestogens and estradiol declined post-exercise, which is relevant because progestogens are precursors of corticosteroids [25].

Sleep duration was also associated with androgen metabolism. In our study, pregnant women with sleep disturbance (higher Pittsburgh Sleep Quality Index) or who reported sleeping more than 7 h in the first trimester showed lower testosterone S/G ratio, indicating a relative reduction in inactive versus active testosterone. Limited evidence in pregnant women exists, but patterns in non-pregnant women support this observation. In a large NHANES study, women aged 41–64 years who reported very short sleep time (<6 h) or long sleep (≥9 h) showed lower testosterone compared to women sleeping 7–8 h [50].

Prenatal tobacco exposure was associated with higher cortisol, cortolone-G, and THE-G in both cohorts. Rizwan et al. (2007) reported that maternal smoking was associated with higher fetal testosterone, supporting the hypothesis that tobacco smoking can alter androgen metabolism in utero [51]. Cajachagua-Torres et al. (2021) found that children prenatally exposed to maternal cannabis combined with tobacco had higher hair cortisol levels at 6 years of age [52].

Alcohol and dietary fat intake showed inconsistent associations with SH metabolism across cohorts. In BiSC, moderate alcohol intake in the second trimester (0.4–1 g/day) was associated with lower 5α-Pregnandiol-3β-sulfate-20α-glucuronide levels, whereas in INMA-Sabadell, for alcohol intake in the first trimester, the association was in the opposite direction. These differences may reflect potential cohort-specific differences in drinking patterns, timing of exposure, or residual confounding. Previous studies indicate that alcohol exposure can reduce estrogen and progestogen levels in fetal alcohol syndrome [27]. In the non-pregnant female population, chronic alcohol exposure suppresses progesterone and its neuroactive metabolites, such as allopregnanolone and isopregnanolone, with levels partially restored during detoxification [53].

Dietary fat intake also showed opposite associations across cohorts. In BiSC, higher energy-adjusted fat intake during the second trimester was positively associated with 5AD-diS_2 and 16OH-DHEAS, suggesting that fat intake may contribute to increased androgens during mid-pregnancy. In contrast, in INMA, where fat intake was assessed in the third trimester, these same metabolites were inversely associated, indicating that the timing of dietary assessment or cohort-specific nutritional patterns could influence the relationship between fat intake and androgen metabolism. Previous studies in pregnant women and non-pregnant adults generally have reported no significant effect of dietary fat on SH [15,54,55].

### 3.4. Maternal Age and Ethnicity as Sociodemographic Determinants Associated with SH Metabolism

In both cohorts, older pregnant women had lower androgen, corticosteroid, and progesterone metabolites and CYP17 activity. Barret et al. (2019) and Kallak et al. (2017) reported that older women had lower serum testosterone levels in the first and third trimester of pregnancy [13,14].

Ethnicity was another consistent determinant of corticosteroid metabolism in both cohorts. Latin American-born pregnant women were more likely to have lower cortolone levels compared to Spanish women. These differences may reflect a combination of biological and psychological factors. Ethnic variation in hypothalamic–pituitary–adrenal axis function has been documented, with African American pregnant women having lower levels of cortisol compared to non-Hispanic white women, and higher adrenocorticotropic hormone levels than Hispanic women [56]. Psychosocial stressors, such as ethnic discrimination and depression risk, can also influence maternal cortisol and its metabolite levels [57,58]; however, this topic was not considered in our analysis.

### 3.5. Genetic: SNPs from Steroidogenesis Enzymes

Common genetic variants in *SRD5A2*, *CYP17A1*, and *CYP19A1* may influence multiple enzymatic steps in steroidogenesis, contributing to interindividual variation in maternal steroid profiles during pregnancy. Although genetic data were not included in the replication analysis due to the high percentage of missing values, some maternal SNPs involved in key steroidogenic enzymes showed associations with specific SH and activity ratios in the BiSC.

Variants in *SRD5A2* (rs2300701, rs806645) were linked to higher levels of androgen and corticosteroid metabolites and 5α-reductase activity. SRD5a2 is a critical enzyme in the metabolism of androgens [59]. The *CYP17A1* rs743572 variant was associated with both androgens and corticosteroids, suggesting modulation of the 17α-hydroxylase and 17,20-lyase. Within *CYP19A1*, rs6493497 and rs700518 were associated with higher progestogen metabolites and lower 11β-HSD2 activity. While the mechanistic link is not fully understood, these results highlight that variation in aromatase, which is involved in androgen-to-estrogen conversion, may be part of interindividual differences in SH metabolism during pregnancy. Kallak et al. (2017) also examined *CYP19A1* rs700518 in pregnant women at 35–39 gestational weeks and found that mothers carrying male fetuses with the CC genotype had higher testosterone levels than carriers of the T allele, consistent with reduced aromatase activity [13].

### 3.6. Fetal Determinants: Sex

Fetal sex contributed to variation in maternal corticosteroid and progestogen metabolism. In both cohorts, pregnant women with female fetuses had higher 3β-HSD enzymatic activity and lower levels of the progesterone metabolite 5PD-20one-S. These findings reinforce the concept that fetal sex modulates maternal endocrine function. Toriola et al. (2011) reported that after the first trimester, maternal progesterone concentrations were about 6% lower in pregnancies with female fetuses compared to male fetuses [29]. Colicino et al. (2023) reported that increased maternal stress was associated with androgen and estrogen levels in pregnant women carrying males with 29.5 weeks of gestation, but no associations were found with corticosteroids and progestogens [60].

### 3.7. Strengths, Limitations, and Future Directions

This study has several notable strengths. First, the analysis was conducted in a well-characterized cohort of 721 Spanish pregnant women in the third trimester, a critical window of hormonal adaptation with relevance for both maternal and fetal outcomes. Second, we employed a high accuracy targeted metabolomic approach of a wide panel of SH metabolites in urine samples, a non-invasive and physiologically relevant biological matrix. Third, our analysis considered a wide range of maternal determinants, including sociodemographic, clinical, nutritional, and genetic variables. Fourth, the replication of associations across two independent cohorts enhances the robustness and generalizability of the results in Spanish pregnant women. Finally, we characterized a general population of pregnant women, not pregnant women with serious complications, which makes this study more generalizable and a base for future clinically oriented studies.

Our study faced some limitations. Although we considered a wide range of maternal determinants, including sociodemographic, clinical, lifestyle, and genetic factors, the models explained less than 20% of the variability in maternal SH levels. This indicates that other relevant contributors were not included in our analysis. Several additional determinants could help explain the remaining variability, such as environmental exposures (air pollution, endocrine-disrupting chemicals, and ambient temperature [61,62,63,64,65]; medications related to lipid metabolism such as statins [1]; and maternal microbiome [66]. Additional unmeasured genetic variants, including those in the fetal genome, may also contribute to interindividual variability in SH metabolism.

Circadian rhythms, which regulate SH secretion and enzymatic activity, can introduce intra-day variation in SH levels that single-timepoint measurements do not capture. In the BiSC, this variability was reduced by analyzing weekly pooled urine samples. In contrast, the replication cohort (INMA–Sabadell) relied on a single spot urine sample, which may have increased variability and partly explained cohort-specific differences.

The cross-sectional design, with SHs measured at a single timepoint in the third trimester, limits causal inference and observation of longitudinal changes in SHs across pregnancy. Furthermore, some determinants, such as diet and alcohol intake, were not assessed at equivalent gestational ages across cohorts, potentially introducing bias due to temporal variations in maternal physiology and lifestyle during pregnancy. Missing data in genetic SNPs, mental health (stress and depression), fertility treatment, medication, PCOS, and some OC were present in the replication cohort. Finally, as both cohorts were based in Spain, the findings may not be fully generalizable to other populations.

Future research should integrate multiple omics layers, including transcriptomic, metabolomic, and microbiome data, together with high-resolution environmental exposure measures within unified analytical frameworks. Longitudinal studies will be essential to clarify causal pathways and capture dynamic SH metabolism during gestation.

## 4. Materials and Methods

### 4.1. Study Participants and Study Overview

In this study, we included data from two Spanish cohorts. The primary analysis was performed in the BiSC, comprising 1080 pregnant women recruited during the first routine prenatal visit (11–15 weeks) at three hospitals in Barcelona, Spain. The replication study used data from the INMA-Sabadell cohort, which recruited 657 pregnant women older than 16 years old, between July 2004 and July 2006, who visited the primary health center of Sabadell for an ultrasound in the 1st trimester. A detailed description of the recruitment process, follow-ups, and data collection are presented elsewhere [67,68]. All participants provided informed consent, and ethics approval was obtained from the corresponding authorities in all the participating institutions and hospitals/medical health centers.

For the present study, we included 721 pregnant women from the BiSC (primary analysis) and 500 from INMA (replication analysis) at 32 weeks of gestation with available data on urinary steroid hormones and with more than 50% of available determinant data. Participants who did not present these criteria were excluded (379 from BiSC and 157 from INMA-Sabadell). The study flow diagram is represented in Figure 2.

### 4.2. Urine Collection and Steroid Targeted Metabolome Profiling

In the BiSC, pregnant women were instructed on how to collect the urine samples on day 1, and fieldworkers brought all the sample collection materials to their homes. Mothers collected their morning and bedtime urine samples from day 2 to day 7 and stored them in their home freezers (around −20 °C). On day 8, urine samples were then collected from the homes by the field workers and taken to the BiSC biobank and kept in −20 °C freezers until the time of lab analysis. The total number of samples was 6 × 2 = 12 urine samples. For the analysis, 0.5 mL pool urine samples with at least 10 voids were selected. In the INMA-Sabadell cohort, urine samples were gathered during the morning interviews as spot samples. Each sample was collected in 100 mL polyethylene containers and promptly stored at −20 °C to maintain stability until the analysis.

SH metabolites were analyzed using LC-MS/MS by Hospital del Mar Medical Research Institute’s Applied Metabolomics Research Laboratory following a validated method described by Servin-Barthet, C. et at. (2025) [41]. On the day of analysis, urine samples were thawed and prepared for extraction of steroid phase II metabolites using solid-phase extraction. Each sample (1 mL) was spiked with 15 μL of internal standards and acidified with 1 mL of 4% aqueous phosphoric acid. SPE was conducted with Oasis HLB cartridges conditioned with methanol, water, and 2% formic acid in water. Steroids were eluted in two stages using methanol and ammonia in methanol. The eluates were dried under nitrogen, reconstituted, combined, and evaporated. The residue was reconstituted in 100 μL of 10% acetonitrile in water for UHPLC-MS/MS analysis. Calibration curves and quality control (QC) samples were prepared for each injection batch.

Chromatographic separation was performed using an Acquity UPLC system with a CSH C18 column. The mobile phases included acetonitrile:water (9:1) and water, both with 25 mM ammonium formate. A gradient elution over 20 min was applied. Mass spectrometric analysis was conducted on a XEVO TQ-S micro mass spectrometer using negative ionization mode with selective reaction monitoring for quantification.

Data was processed with MassLynx (Waters Corporation, Milford, MA, USA) and TargetLynx (Waters Corporation, Milford, MA, USA) software. Analyte responses were calculated as ratios of analyte to internal standard areas. Concentrations of endogenous steroid metabolites were determined using calibration curves in stripped urine. Metabolites without quantitative standards were relatively quantified using structurally similar compounds. QC samples were included in each of four analytical batches. Limits of detection (LODs) ranged approximately from 1 to 200 ng/mL, with most androgens and estrogens falling below 10 ng/mL and higher LODs observed for some progestogens and precursors. Features with below limit of detection were log transformed and values below LOD were imputed using left-censored data [69]. Of the 57 steroid hormones detected in urine, 7 with poor reproducibility (RSD > 35%) were excluded, resulting in 50 hormones included in the analyses (Appendix A). Based on 50 individual SH metabolite concentrations, transformed into µmol/L, we estimated the sum of 13 steroid family or subfamilies, 9 sulfate/glucuronide (S/G) ratio, and 17 steroid hormone enzymatic activity markers. Enzymatic phase I activity was estimated using the ratios of molar concentration product/precursor. Product/precursor ratios can explain physiology implications of steroid hormones production [70,71]. A total of 89 steroidomic outcomes were analyzed, including individual SH and derived metabolic indicators. To standardize the SH metabolite data, urine specific gravity (SG) was measured in specimens using the Digital Urine Specific Gravity Refractometer (ATAGO CO., LTD, Tokyo, Japan) to correct for dilution using metabolite-specific regressions to remove SG dependency while preserving biological variation. All SH concentrations, including 50 individual SH and 14 sum of SH families, were corrected by SG and expressed as µmol/L. Enzymatic ratios and sulfate/glucuronide ratios were estimated based on raw concentration of steroid hormones (µmol/L) without adjustment by specific gravity. Finally, SH metabolites were corrected by gestational age using the MoM multiples of the normal median (MoM) approach based on quantile regression with the objective of measuring how far an individual test result deviates from the median of the same gestational age [72].

### 4.3. Determinants

#### 4.3.1. Sociodemographic and Clinical Parameters

For the BiSC, sociodemographic data such as age, country of origin, and education levels were collected through the face-to-face interview during hospital visits [67]. Data on clinical and reproductive history included gynecological conditions, fertility treatment, parity, contraceptive methods. The clinical data of pregnancy including clinical examinations and pregnancy complications (gestational diabetes, preeclampsia, and abnormal amniotic fluid) were obtained from the hospital records and self- administered questionnaires. Anthropometric parameters including maternal weight and height were obtained by physical examination of mothers at hospital visits in the first and third trimesters, which were then used to calculate BMI, measured in kg/m^2^.

For INMA-Sabadell cohort, sociodemographic (maternal age, education, and ethnicity) and clinical parameters (obstetric complications and parity) were collected from self-report questionnaires [68]. Maternal pre-pregnancy BMI was calculated from self-reported weight and height measured in the first trimester. Gestational age was estimated based on the date of the last menstrual period.

#### 4.3.2. Fetal Sex and Growth

For the BiSC and INMA cohorts, fetal growth and sex were obtained by ultrasonographic measurements carried out by trained sonographers. Fetal weight for the first and third trimester of gestation was estimated using the Hadlock formula [73]. Estimated fetal weight was standardized following the guideline by International Society of Ultrasound in Obstetrics and Gynecology (ISUOG) [73,74] for BiSC and, in the case of INMA-Sabadell cohort, standardized fetal weight was estimated following the method described by Iñiguez et al. (2015) [75].

#### 4.3.3. Genetic

Genetic polymorphism data were only analyzed for the BiSC, since over 70% of the corresponding data were missing in the INMA cohort. Peripheral blood was collected from mothers in EDTA tubes during pregnancy (12 or 32 weeks) or at delivery, and maternal DNA was extracted from samples randomly selected for analysis. EDTA tubes were centrifuged at 2000× *g* for 10 min and plasma; buffy coat and red cells were separated. DNA was extracted from 200 ul of buffy coat using the QIAsymphony platform and the QIAsymphony DSP DNA Mini Kit at the Hospital del Mar Research Institute (IMIN). Genome-wide genotyping was performed using the Infinium Global Screening Array v3.0 with Multi-Disease content (Illumina, 730,059 variants, San Diego, CA, USA). DNA was quantified using the Quant-iT™ PicoGreen™ kit. Genotyping followed the manufacturer’s protocols, with HapMap controls included on each plate. Genotype calling and clustering were performed using GenomeStudio Gen Train 3.0, and variants were annotated in b37 + strad using the GSAMD-24v3-0-EA_20034606_A1 manifest. In the genetic quality control data, variants with a call rate < 97% or MAD < 1% or which failed the Hardy–Weinberg equilibrium (*p* < 1 × 10^−6^) were excluded from the analysis. Relatedness was assessed using PI_HAT, and 54 s-degree relatives (PI_HAT > 0.25) were excluded. Principal components (PCs) were estimated on pruned SNPs; the first 20 PCs explained 72.2% of the population variance and 35.4% of the variance in Europeans. Imputation was conducted using the HRC v1.1 reference panel via the Sanger Imputation Service, employing EAGLE2 for phasing and PBWT for imputation. Post-imputation filters included MAD > 1%, HWE *p* > 1 × 10^−6^, and INFO > 0.8, resulting in approximately 5 million high-quality SNPs. From the imputed dataset, we selected SNPs previously associated with steroid hormone enzymatic pathways described in studies conducted on pregnant and non-pregnant adults: CYP3A4 (rs2300701, rs806645), CYP1B1 (rs1056836), HSD3B1 (rs1264459), HSD17B2 (rs1937863), CYP17A1 (rs743572), and CYP19A1 (rs10046, rs28757184, rs700518, rs6493497) [13,22,76].

#### 4.3.4. Mental Health

Mental health determinants (stress and depression) were only available in the BiSC and was considered in the data from the third trimester of gestation using validated questionnaires [67]. The Edinburgh Postnatal Depression Scale was used to evaluate peripartum depression [77]. Maternal stress was evaluated using the 10-item Perceived Stress Scale (PSS−10) filled by the mothers [78,79].

#### 4.3.5. Lifestyle

Lifestyle determinants were collected in the first and/or third trimester in both cohorts.

From the BiSC, physical activity data was obtained through the Pregnancy Physical Activity Questionnaire [80] filled out by the participants in the first and third trimesters of pregnancy. Physical activity of moderate or vigorous intensity was considered as a determinant in the study and was categorized according to the physical activity WHO recommendation into more/less than 150 min/week [81]. Sleep-related variables in the first and third trimesters of pregnancy were estimated as the difference in hours between self-reported bedtime and wake-up time, and sleep quality was characterized through the Pittsburgh Sleep Quality Index [82]. Data on smoking and alcohol consumption during pregnancy were obtained from self-administered validated questionnaires. Maternal diet data was collected using a validated food frequency questionnaire (FFQ) during the second trimester of pregnancy [83]. Mediterranean diet (MED) score (not including alcohol) was obtained from the FFQ data [84]. Nutrient intake was estimated from US and Spanish food composition tables [85,86] and was adjusted for energy intake following the energy-residual method described by Willet et al. (1997) [87].

In the INMA-Sabadell cohort, smoking and alcohol intake during pregnancy was obtained through self-reported questionnaires. Dietary pattern and adherence to relative MED (rMED) were assessed using a validated FFQ. Nutrient intake was estimated from the US and Spanish food composition tables [86,88]. Physical activity was measured using a validated questionnaire and expressed as metabolic equivalent (MET) units.

### 4.4. Statistical Analysis

Values for means, medians, and standard deviation (SD) were calculated for continuous variables and frequencies were calculated for categorical variables. The descriptive analysis was performed using no imputed data.

#### 4.4.1. Data Preparation

We performed data selection steps, from the initial selection of data to the filtering of strongly correlated and noisy data in both cohorts.

First, we used a wide selection of determinants associated with steroid hormones based on the existing literature (Appendix A). To reduce the amount of missingness in the data, we discarded records of individuals with more than 50% of missing data in all the variables (5 participants were discarded in BiSC and 0 in INMA-Sabadell) and variables with more than 50% missing data (all SNPs (*n* = 10) were discarded in INMA-Sabadell). The percentage of participants with missing values of determinant was 0.68% for BiSC and 0% for INMA-Sabadell. Then, we further refined the selection of determinants by removing highly correlated variables (correlation coefficient r > 0.9). As a result, we excluded one variable (BMI in the first trimester) from BiSC, while no variables were excluded from INMA-Sabadell. For the statistical analysis, missing values were imputed using the missForest R package, a single interactive imputation algorithm that can handle both categorical and continuous variables and capture nonlinear relationships. The full description of the preselection steps is available in Appendix A, and a detailed list of names of variables selected in both cohorts is available in Appendix A.

A total of 47 determinants from the BiSC and 28 determinants from the INMA-Sabadell cohort were considered in the analysis (Figure 8, Appendix A). Determinants covered sociodemographic (age, ethnicity, and education levels), clinical parameters (clinical and reproductive history), OC, BMI, lifestyle factors (diet, supplements, physical activity, sleep, alcohol intake, smoking), mental health indicators (depression and stress in the 3rd trimester), selected maternal SNPs, and fetus determinants (sex and estimated weight in the first/second and in the third trimester). In the replication cohort (INMA-Sabadell), the following data was not available or more than 70% of it comprised missing data: SNPs, mental health (stress and depression), fertility treatment, medication, PCOS and some OC like preeclampsia, and type I diabetes.

#### 4.4.2. Primary Analysis—BiSC

##### ExWAS Analysis

Interquartile range (IQR) standardization of the continuous predictors was applied to normalize data distributions and mitigate the influence of outliers. Prior to fitting the model, all outcomes were log2-transformed to approximate normality. ENT was applied to correct multiple comparisons *P* (*P_ENT_*). *P_ENT_* = 0.05/(ENT metabolitesxENT determinants). A different ENT was applied for each group of outcomes: SH metabolites, sum of SH, S/G ratios, and phase I enzymatic activity. *P_ENT_* for SH metabolites was 0.00006, the sum was 0.0002, S/G ratio was 0.0003, phase I enzymatic activity was 0.0001.

##### Elastic Net Regression Analysis to Determinant Selection

Elastic net (ENET) regression was employed to identify the associations between a set of determinants and hormone levels in the BiSC. ENET, a regularized regression technique that combines both Lasso (L1) and Ridge (L2) penalties, was selected due to its ability to handle multicollinearity and high-dimensional data while reducing overfitting [89]. ENET regularization parameter α was set to 0.5 to balance the contribution of Lasso and Ridge penalties. The optimal regularization parameter λ was determined through 10-fold cross-validation using the “mean squared error” as the performance metric. Specifically, we selected the minimum λ value corresponding to the highest penalty. Preprocessing of predictors involved standardizing continuous variables through IQR scaling to normalize data distributions and mitigate the influence of outliers. Categorical variables were converted to dummy variables to ensure proper inclusion in the model. Each outcome variable was modeled separately using a train–test split (80–20%) to facilitate robust model evaluation and ensure that results could be validated against an independent dataset. ENET was applied to assess the relationships between the clinical, genetic, and environmental predictors and hormonal outcomes. A set of covariates (season of birth, hospital, and COVID-19 confinement status) was forced into the model by setting their penalty factors to zero. Non-zero coefficients from the ENET model were interpreted as the key predictors of hormone levels. Additionally, to assess model stability, we performed 100 bootstrap replications of resampling. For each outcome, determinants that were selected in at least 80% of the replication of ENET analyses were considered stable. All analyses were conducted using R (version 4.4.1) and the following packages: glmnet for ENET regression and caret for model training and cross-validation.

##### Linear Regression Model and Variance Explained Estimation

To estimate the best prediction for each SH metabolome outcome, a linear regression model was applied between all the predictors selected by ENET and each outcome, and adjusted by season of birth, hospital, and COVID-19 confinement status in the BiSC. To quantify the contribution of each determinant to the explained variance of steroid metabolome outcomes, we used the relaimpo R package (v2.2.7). Specifically, we applied the lmg method in the calc.relimp function, which decomposes the total model *R*2 into the relative contributions of each predictor by averaging sequential sums of squares over all possible orderings of variables [90]. This provides the percentage of variance explained attributable to each determinant.

#### 4.4.3. Replication Analysis—INMA-Sabadell Cohort

Determinants identified in BiSC via ENET were evaluated in INMA-Sabadell using the same linear regression framework, adjusting for cohort-specific covariates related to urine data collection like season and time of urine collection.

## 5. Conclusions

This study provides a comprehensive and novel characterization of maternal SH metabolism and its determinants during late pregnancy. Using mass spectrometry-based metabolomics, penalized regression models, and external replication, we identified that clinical, lifestyle, and sociodemographic determinants contribute to interindividual variability in the SH metabolome. Our findings highlight the complexity of SH metabolism in later pregnancy and reveal consistent associations with modifiable determinants such as maternal BMI, dietary fat intake, alcohol intake, smoking, physical activity, and sleep and non-modifiable determinants like parity, ethnicity, and sex of the fetus in two independent Spanish cohorts. However, these factors explained less than 20% of the total variability, indicating that approximately 80% remains unexplained and may be influenced by other unmeasured determinants. The integration of such metabolic phenotyping into maternal health research may help identify early biomarkers of pregnancy complications and developmental risk and supports the case for including urinary SH profiling in future longitudinal and clinical studies.

## Figures and Tables

**Figure 1 ijms-26-11598-f001:**
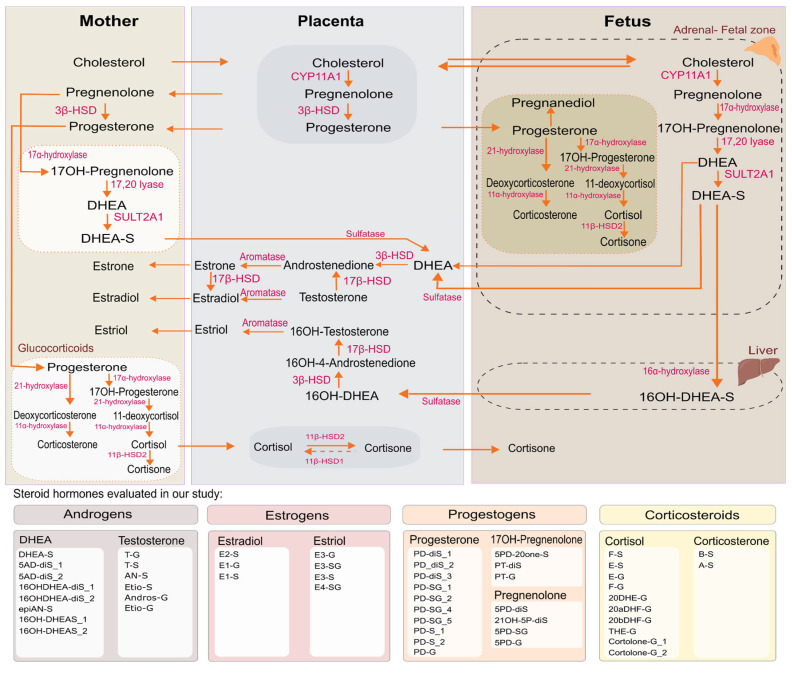
Schematic representation of the steroidogenesis pathways during pregnancy that were studied. The following abbreviations stand for the following compounds: DHEA-S, Dehydroepiandrosterone (DHEA)-sulfate; 5AD-diS_1, 5-androsten-3β17β-diol-disulfate; 5AD-diS_2, 5-androsten-3α17β-diol-disulfate; 16OHDHEA-diS_1, 16β-hydroxy-DHEA-disulfate; 16OHDHEA-diS_2, 16α-hydroxy-DHEA-disulfate; epiAN-S, epiandrosterone-sulfate; 16OH-DHEAS_1, 16β-hydroxy-DHEA-sulfate; 16OH-DHEAS_2, 16α-hydroxy-DHEA-sulfate; T-G, Testosterone-glucuronide; T-S, Testosterone-sulfate; AN-S, Androsterone-sulfate; Etio-S, Etiocholanolone-sulfate; Andros-G, Androsterone-glucuronide; Etio-G, Etiocholanolone-glucuronide; B-S, Corticosterone-sulfate; A-S, 11-dehydrocorticosterone-sulfate; F-S, Cortisol-sulfate; E-S, Cortisone-sulfate; E-G, Cortisone-glucuronide; F-G, Cortisol-glucuronide; 20DHE-G, 20dihydrocortisone-glucuronide; 20aDHF-G, 20α-dihydrocortisol-glucuronide; 20bDHF-G, 20β-dihydrocortisol-glucuronide; THE-G, Tetrahydrocortisone-glucuronide; Cortolone-G_1, 20α-Cortolone-glucuronide; Cortolone-G_2, 20b-Cortolone-glucuronide; E2-S, Estradiol-Sulfate; E1-G, Estrone-glucuronide; E1-S, Estrone-sulfate; E3-G, Estriol-glucuronide; E3-SG, Estriol-sulfoglucoconjugated; E3-S, Estriol-sulfate; E4-SG, Estetrol-sulfoglucoconjugated; 5PD-20one-S, 17-hydroxy-5-pregnenolone-3-sulfate; PT-diS, Pregnantriol-disulfate; PT-G, Pregnantriol-glucuronide; 5PD-diS, 5-Pregnendiol-disulfate; 21OH-5P-diS, 21-Hydroxypregnenolone-disulfate; 5PD-SG, 5-Pregnendiol-sulfoglucoconjugated; 5PD-G, 5-Pregnendiol-glucuronide; PD-diS_1, 5α-Pregnan-3β,20α-diol-diSulfate; PD_diS_2, 5α-Pregnan-3α,20α-diol-disulfate; PD-diS_3, 5β-Pregnan-3α,20α-diol-disulfate; PD-SG_1, 5α-Pregnandiol-3β-sulfate-20α-glucuronide; PD-SG_2, Pregnandiol-sulfoglucoconjugate; PD-SG_4, Pregnandiol-sulfoglucoconjugate; PD-SG_5, Pregnandiol-sulfoglucoconjugate; PD-S_1, 5α-Pregnan-3β20α-diol-20-sulfate; PD-S_2, 5β-Pregnan-3α20α-diol-20-sulfate; and PD-G, Pregnandiol-glucuronide.

**Figure 2 ijms-26-11598-f002:**
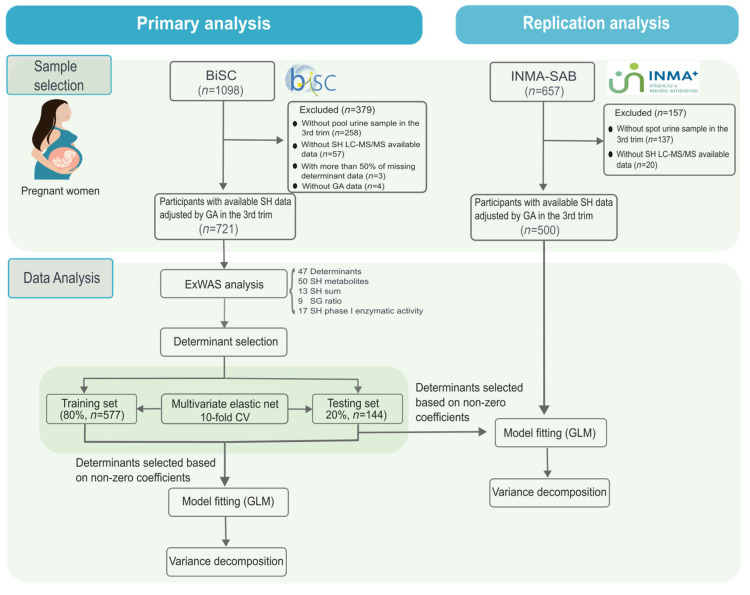
Flow diagram of participant selection and data analysis.

**Figure 3 ijms-26-11598-f003:**
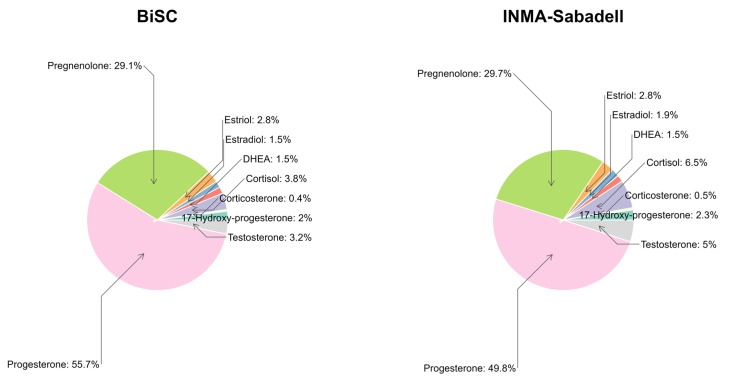
Urinary steroid hormone concentration proportion by family from BiSC (*n* = 721) and INMA-Sabadell cohort (*n* = 500). DHEA, dehydroepiandrosterone.

**Figure 4 ijms-26-11598-f004:**
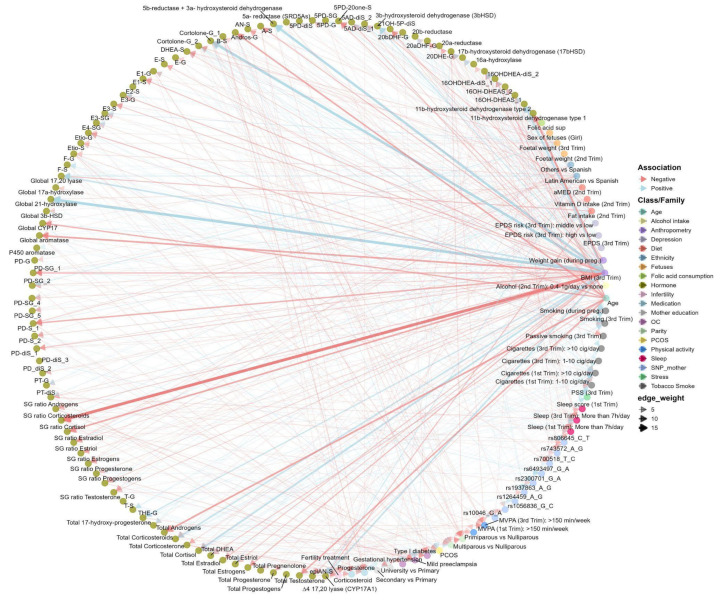
Network of significant associations between SH metabolome and main determinants in the BiSC (*n* = 721). The main determinants included physiological factors (mother and fetus), sociodemographic variables, genetics (polymorphism of steroid hormones enzymes), medical history, stress, depression, and lifestyle factors (alcohol intake, smoking, dietary intake, physical activity, and sleep pattern). Determinants were normalized using the interquartile range (IQR). The SH metabolome was log2-transformed. All associations were adjusted for potential confounders, including hospital of delivery in the 3rd trimester or at birth, COVID-19 exposure period, and season of birth. The SH metabolome included individual hormones, hormone groups, and enzymatic activities. Associations with *p*-values below 0.05 were considered significant. Edge weights represent -log10 (*p*-value), so thicker edges indicate stronger associations.

**Figure 5 ijms-26-11598-f005:**
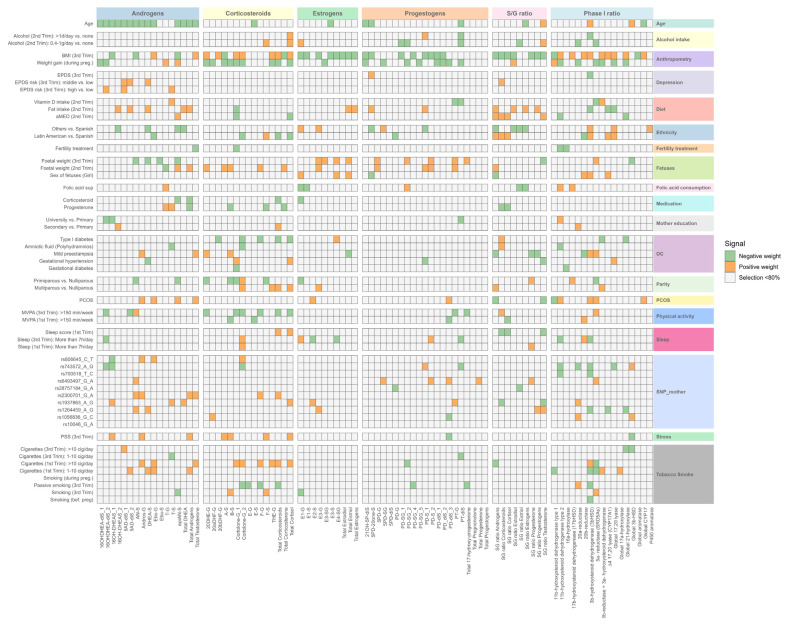
Variable selection in elastic net regression for SH metabolome from the BiSC. Heatmap illustrating the selection of variables in ENET regression for steroid hormone metabolites from the BiSC. Rows correspond to determinants grouped by family categories, and columns correspond to SH metabolites grouped into androgens, corticosteroids, estrogens, progestogens, S/G ratio, and Phase I enzymatic activity. Tile color shows coefficient direction (positive: soft orange; negative: soft blue), while very light gray indicates variables selected in less than 80% of 100 bootstrap replications.

**Figure 6 ijms-26-11598-f006:**
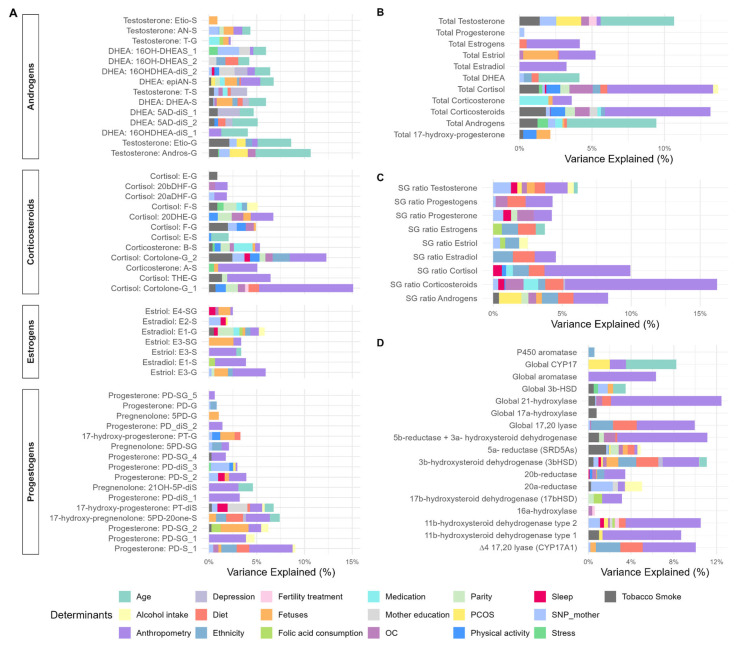
Variance explained between main determinants selected by ENET and steroid hormone metabolome from BiSC (*n* = 721). Panels: (**A**) individual steroid hormone metabolites (49), (**B**) sum of steroid hormones (11), (**C**) sulfate-to-glucuronide ratio (9), (**D**) enzymatic activity (product-to-precursor ratio) (17). Variance explained (R^2^) estimated from linear regression models with ENET-selected determinants, adjusted for hospital at third trimester or birth, COVID-19 exposure period, and season. OC, obstetric complication; PCOS, Polycystic Ovary Syndrome. Anthropometry variables included weight gain during pregnancy (from 1st trimester to 3rd trimester) and body mass index in the 3rd trimester. OC included diabetes, preeclampsia, hypertension and abnormal amniotic fluid. SNIP included rs1056836, rs1264459, rs1937863, rs2300701, rs28757184, rs6493497, rs700518, rs743572, rs806645.

**Figure 7 ijms-26-11598-f007:**
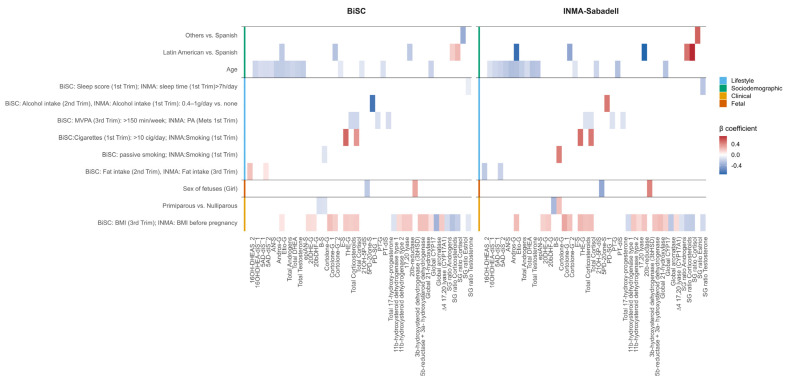
Significant and consistent associations between determinants and steroid hormone (SH) molecular features in BiSC (*n* = 721) and INMA-Sabadell (*n* = 500) cohorts. BMI body mass index, MVPA moderate–vigorous physical activity, S/G sulfate/glucuronide ratio, Andros-G Androsterone-glucuronide, AN-S Androsterone-sulfate, 5AD-diS_1 5-androsten-3β17β-diol-disulfate, 5AD-diS_2 5-androsten-3α17β-diol-disulfate, B-S Corticosterone-sulfate, Cortolone-G Cortolone-glucuronide, Cortolone-G_1 20α-Cortolone-glucuronide, Cortolone-G_2 20b-Cortolone-glucuronide, 20bDHF-G 20β-dihydrocortisol-glucuronide, 20DHE-G 20dihydrocortisone-glucuronide, E-S Cortisone-sulfate, epiAN-S Epiandrosterone-sulfate, Etio-G Etiocholano-lone-glucuronide, 16OH-DHEAS_2 16α-Hydroxy-DHEA-sulfate, 16OHDHEA-diS_1 16β-Hydroxy-DHEA-disulfate, 21OH-5P-diS 21-Hydroxypregnenolon-disulfate, PD-SG_1 5α-Pregnan-3β20α-diol-20-sulfate, 5PD-20one-S 17-Hydroxy-5-pregnenolone-3-sulfate, PT-G Pregnantriol-glucuronide, PT-diS Pregnantriol-disulfate, and THE-G Tetrahydrocortisone-glucuronide. In parity, cortolone-G is equivalent to Cortolone-G1 in BiSC and Cortolone-G2 in INMA-Sabadell. In tobacco smoking, cortolone-G is equivalent to cortolone-G2 in BiSC and Cortolone-G1 in INMA-Sabadell.

**Figure 8 ijms-26-11598-f008:**
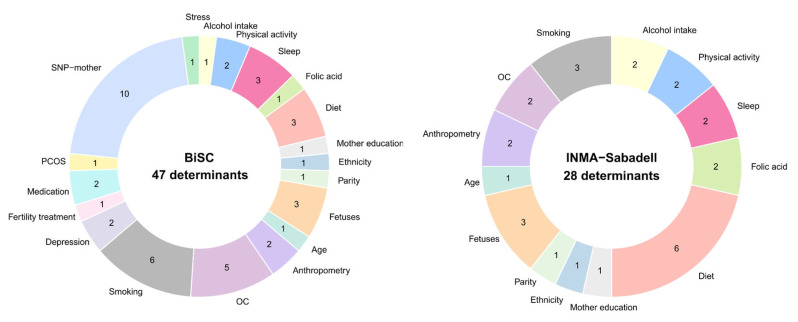
Determinants considered analyzed in BiSC (*n* = 719) and INMA-Sabadell (*n* = 500) cohorts. OC, obstetric complications; PCOS, Polycystic Ovary Syndrome.

**Table 1 ijms-26-11598-t001:** Baseline characteristics of participants.

	BiSC (*n* = 721)	INMA-Sabadell (*n* = 500)
Maternal age (year)	34.8 [5.2] ^1^	31.2 [5.7] ^1^
*Missing*	*0*	*1*
Gestational age at sampling collection	34.7 [2] ^1^	34.1 [1.4] ^1^
BMI * (kg/m^2^)	23.2 [4.9] ^1^	22.7 [4.5] ^1^
Underweight (<18.5 kg/m^2^), *n* (%)	22 (4)	25 (5)
Normal weight (18.5–25 kg/m^2^) *n* (%)	434 (64)	331 (66)
Overweight (25–30 kg/m^2^), *n* (%)	155 (23)	102 (20)
Obese (>30 kg/m^2^), *n* (%)	63 (9)	42 (8)
* Missing*	*47*	*0*
Mother country of birth, *n* (%)		
Spain	498 (69)	446 (89)
Latin America	165 (23)	36 (7)
Others	57 (8)	18 (4)
* Missing*	*1*	*0*
Maternal education, *n* (%)		
Primary or without education	27 (4)	4 (1)
Secondary	175 (24)	132 (27)
Technical or University	519 (72)	362 (73)
* Missing*	*1*	*2*
Parity, *n* (%)		
Nulliparous	443 (61)	282 (57)
Primiparous	229 (32)	185 (37)
Multiparous	49 (7)	31 (6)
* Missing*	*0*	*2*
Alcohol during pregnancy (daily intake), *n* (%)		
0.4–1 g/day	18 (3)	33 (7)
1 g/day or more	13 (2)	36 (7)
None	593 (95)	429 (86)
* Missing*	*97*	*2*
Smoking at the beginning of pregnancy, *n* (%)		
between 1 and 10 cig/day	10 (2)	48 (10)
more than 10 cig/day	15 (2)	19 (4)
No smoking	684 (96)	427 (86)
* Missing*	*12*	*6*
Physical activity, *n* (%)		
More than 150 min/week	151 (22%)	*-*
* Missing*	*21*	-
Physical activity (mets)	-	37.5 [5.6]
* Missing*	-	*2*
Sleep time, *n* (%)		
More than 7 h/day	554 (79%)	431 (87%)
* Missing*	*20*	*2*
Folic acid supplementation, *n* (%)		
Yes	380 (74%)	408 (82%)
* Missing*	*209*	*0*
Mediterranean Diet Score	4.0 [2.0]	4.0 [2.0]
* Missing*	*97*	*6*
Contraceptive, *n* (%)		
Yes	627 (87%)	452 (90%)
* Missing*	*8*	*1 (0.2%)*
Type of contraceptive, *n* (%)		
Hormonal	145 (20%)	175 (35%)
Season, *n* (%)		
Autumn	179 (25)	139 (28)
Spring	175 (24)	139 (28)
Summer	214 (30)	107 (22)
Winter	153 (21)	111 (22)
* Missing*	-	4
COVID confinement, *n* (%)		-
Yes	194 (27)	-

^1^ Median (IQR); n (%). * Maternal body mass index (BMI) in INMA-Sabadell was related to pre-pregnancy BMI and in BiSC was BMI in the first trimester. Alcohol during pregnancy in INMA-Sabadell was measured in the first trimester and in BiSC in the second trimester. The type of contraception prior to the last pregnancy was classified as hormonal or non-hormonal. In BiSC, hormonal contraceptive methods were defined as the use of birth control pills, vaginal ring, transdermal patch, hormonal intrauterine device (IUD), prolonged-acting progesterone injection, or subcutaneous hormonal implant; in INMA, they were defined as birth control pill, injection, or IUD.

**Table 2 ijms-26-11598-t002:** Concentration of steroid hormones in participants from BiSC and INMA-Sabadell.

	BiSC (*n* = 721)	INMA (*n* = 500)
Hormone (µmol/L)	Median (IQR)	Median (IQR)
17-hydroxy-pregnenolone: 5PD-20one-S	1.99 (2.64)	2.75 (3.85)
17-hydroxy-progesterone: PT-G	3.27 (2.09)	4.30 (3.84)
17-hydroxy-progesterone: PT-diS	0.05 (0.07)	0.07 (0.12)
Corticosterone: A-S	0.30 (0.20)	0.35 (0.46)
Corticosterone: B-S	0.39 (0.30)	0.59 (0.70)
Cortisol: 20DHE-G	0.10 (0.10)	0.11 (0.15)
Cortisol: 20αDHF-G	0.03 (0.04)	0.07 (0.08)
Cortisol: 20βDHF-G	0.31 (0.30)	0.33 (0.48)
Cortisol: Cortolone-G_1	2.33 (1.76)	3.85 (4.31)
Cortisol: Cortolone-G_2	1.08 (0.86)	2.50 (2.96)
Cortisol: E-G	0.33 (0.22)	0.58 (0.69)
Cortisol: E-S	0.03 (0.02)	0.04 (0.04)
Cortisol: F-G	0.08 (0.06)	0.22 (0.21)
Cortisol: F-S	0.18 (0.12)	0.28 (0.30)
Cortisol: THE-G	1.74 (1.46)	3.56 (3.78)
DHEA: 16OH-DHEAS_1	0.87 (0.65)	0.63 (0.86)
DHEA: 16OH-DHEAS_2	0.60 (0.85)	0.85 (1.44)
DHEA: 16OHDHEA-diS_1	0.34 (0.38)	0.39 (0.51)
DHEA: 16OHDHEA-diS_2	0.12 (0.13)	0.14 (0.21)
DHEA: 5AD-diS_1	0.04 (0.04)	0.05 (0.07)
DHEA: 5AD-diS_2	0.11 (0.10)	0.16 (0.20)
DHEA: DHEA-S	0.12 (0.33)	0.26 (0.97)
DHEA: epiAN-S	0.07 (0.14)	0.13 (0.26)
Estradiol: E1-G	0.73 (0.70)	1.12 (1.25)
Estradiol: E1-S	1.59 (3.22)	2.22 (4.89)
Estradiol: E2-S	0.02 (0.03)	0.03 (0.05)
Estriol: E3-G	2.69 (1.82)	2.93 (2.18)
Estriol: E3-S	0.76 (0.90)	1.02 (1.31)
Estriol: E3-SG	1.05 (1.22)	1.19 (1.46)
Estriol: E4-SG	0.03 (0.04)	0.03 (0.05)
Pregnenolone: 21OH-5P-diS	0.20 (0.15)	0.28 (0.24)
Pregnenolone: 5PD-G	45.04 (30.24)	52.53 (48.78)
Pregnenolone: 5PD-SG	0.47 (0.44)	0.52 (0.62)
Pregnenolone: 5PD-diS	2.46 (1.80)	3.17 (2.95)
Progesterone: PD-G	72.62 (43.54)	76.60 (57.04)
Progesterone: PD-SG_1	1.93 (2.27)	1.64 (2.07)
Progesterone: PD-SG_2	0.95 (1.36)	0.68 (1.17)
Progesterone: PD-SG_4	2.66 (2.46)	2.82 (3.15)
Progesterone: PD-SG_5	1.09 (0.98)	1.14 (1.20)
Progesterone: PD-S_1	0.61 (0.73)	0.77 (1.22)
Progesterone: PD-S_2	6.06 (7.89)	7.06 (11.94)
Progesterone: PD-diS_1	1.61 (1.05)	1.75 (1.28)
Progesterone: PD-diS_3	0.04 (0.04)	0.07 (0.09)
Progesterone: PD_diS_2	0.43 (0.41)	0.52 (0.67)
Testosterone: AN-S	0.46 (0.63)	0.83 (1.29)
Testosterone: Andros-G	3.25 (2.63)	5.62 (6.00)
Testosterone: Etio-G	1.09 (1.12)	2.27 (2.82)
Testosterone: Etio-S	0.36 (0.45)	0.78 (1.09)
Testosterone: T-G	0.01 (0.01)	0.02 (0.02)
Testosterone: T-S	0.02 (0.02)	0.02 (0.03)

Values expressed as median (IQR). DHEA-S, Dehydroepiandrosterone (DHEA)-sulfate; 5AD-diS_1, 5-androsten-3β17β-diol-disulfate; 5AD-diS_2, 5-androsten-3α17β-diol-disulfate; 16OHDHEA-diS_1, 16β-hydroxy-DHEA-disulfate; 16OHDHEA-diS_2, 16α-hydroxy-DHEA-disulfate; epiAN-S, epiandrosterone-sulfate; 16OH-DHEAS_1, 16β-hydroxy-DHEA-sulfate; 16OH-DHEAS_2, 16α-hydroxy-DHEA-sulfate; T-G, Testosterone-glucuronide; T-S, Testosterone-sulfate; AN-S, Androsterone-sulfate; Etio-S, Etiocholanolone-sulfate; Andros-G, Androsterone-glucuronide; Etio-G, Etiocholanolone-glucuronide; B-S, Corticosterone-sulfate; A-S, 11-dehydrocorticosterone-sulfate; F-S, Cortisol-sulfate; E-S, Cortisone-sulfate; E-G, Cortisone-glucuronide; F-G, Cortisol-glucuronide; 20DHE-G, 20dihydrocortisone-glucuronide; 20aDHF-G, 20α-dihydrocortisol-glucuronide; 20bDHF-G, 20β-dihydrocortisol-glucuronide; THE-G, Tetrahydrocortisone-glucuronide; Cortolone-G_1, 20α-Cortolone-glucuronide; Cortolone-G_2, 20b-Cortolone-glucuronide; E2-S, Estradiol-Sulfate; E1-G, Estrone-glucuronide; E1-S, Estrone-sulfate; E3-G, Estriol-glucuronide; E3-SG, Estriol-sulfoglucoconjugated; E3-S, Estriol-sulfate; E4-SG, Estetrol-sulfoglucoconjugated; 5PD-20one-S, 17-hydroxy-5-pregnenolone-3-sulfate; PT-diS, Pregnantriol-disulfate; PT-G, Pregnantriol-glucuronide; 5PD-diS, 5-Pregnendiol-disulfate; 21OH-5P-diS, 21-Hydroxypregnenolone-disulfate; 5PD-SG, 5-Pregnendiol-sulfoglucoconjugated; 5PD-G, 5-Pregnendiol-glucuronide; PD-diS_1, 5α-Pregnan-3β,20α-diol-diSulfate; PD_diS_2, 5α-Pregnan-3α,20α-diol-disulfate; PD-diS_3, 5β-Pregnan-3α,20α-diol-disulfate; PD-SG_1, 5α-Pregnandiol-3β-sulfate-20α-glucuronide; PD-SG_2, Pregnandiol-sulfoglucoconjugate; PD-SG_4, Pregnandiol-sulfoglucoconjugate; PD-SG_5, Pregnandiol-sulfoglucoconjugate; PD-S_1, 5α-Pregnan-3β20α-diol-20-sulfate; PD-S_2, 5β-Pregnan-3α20α-diol-20-sulfate; PD-G, Pregnandiol-glucuronide.

## Data Availability

The raw data supporting the conclusions of this article will be available by the authors on request. The script of the main analysis is available in https://github.com/Elaveriano/pregnancy-steroid-determinants (accessed on 20 September 2025).

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
