# Peer review of "Mapping of Determinants of Urinary Sex Steroid Metabolites During Late Pregnancy: Results from Two Spanish Cohorts"

_ijms, 2025, doi:10.3390/ijms262311598_

Round 1
Reviewer 1 Report
Comments and Suggestions for Authors
Dear Colleagues,
Thank you very much for the interesting study you have submitted for publication. There is a lot of data presented here; however, I have some observations and suggestions that I describe below:
Major comments:
- The main aim and the take home message are not precisely characterised. The abstract lacks this information and introduction contains a bit vague description since it is unusual to set metabolite characterization and analysis as a main goal of the study even when you then continue with information about identification of determinants – I would suggest to rewrite it.
- Abstract is mostly focused on methods but more results and conclusion would be beneficial – in this form it seems like no novelty was brought
- I would recommend the introduction to be shortened – it contains a lot of redundant information and should be more addressed. It is questionable if the Table 1 is necessary and it would be beneficial if the information contained would be integrated into the text. Moreover, some facts e.g., vitamin A intake, Black women ethnicity do not serve for the further analysis.
- In the Methods section, inclusion and exclusion criteria for both cohorts are missing. Authors do not describe if participants had some pathologies etc. At first, it is not clear which cohort is used for what kind of analysis with which determinants since this information is a part of the Results but definitely it should be included into Methods. I highly recommend to add the majority of the subchapter 2.1. Participant´s characteristic into methods because it brings characteristics (as named in the title) and not results. Moreover, Table 2 is missing some characteristics used further in the study like vitamin supplements, diet, sleep, contraceptive etc.
- Is the selection of SNPs based only on their relation to steroidogenesis? If so, is there a deeper significance, and are they related to, or candidates for, specific pathologies?
- Regarding results, in the section 2.5 p values and a reference to a table should be added into some sections. E.g. “Pregnant women with PCOS diagnosis before pregnancy had higher androgens levels.”, “Regarding maternal education, women with university level studies had lower progesterone metabolite levels but higher total corticosteroid levels compared with women with primary education”. In general, throughout the Results sections, p values are missing
- I am not sure if information from different trimesters from two cohorts are even comparable (L 375).
- The entire analysis was performed on an impressive BiSC cohort. However, analysis performed on INMA-Sabadell cohort is redundant and does not offer any additional characteristic, contrarily shows opposite results in some ways. What exactly did you want to prove and demonstrate when you did not use the same analysis (Direct replication)? In my opinion, if one wants to replicate a study and calls it a replication analysis it then needs even bigger n, more factors and some add-ons/new parameters which should be added to whole examination and make it more valuable (more conceptual replication). In this case, it was the exact opposite: a smaller sample size and fewer determinants were used, with questionnaires collected at a different time (a different trimester), and different urine samples were taken. Overall, this gives the impression that these might be leftover data from some study, which are ultimately being used as an addendum to another study.
- The authors describe that approximately 80% of SH variability remains unexplained. There may be several factors contributing to this fact, which should be discussed in detail but are not - some are only mentioned in the limitations section. Key aspects that could be addressed include the cumulative effect of various determinants and the influence of pre-pregnancy life.
- Despite the authors use of a novel approach for SH detection, the effects of common determinants (age, BMI, etc.) are already intensively discussed. Thus, the added novelty should be more emphasized.
Discussion section:
- More studies should be used for discussion of the obtained results. In the L 425 is written “This analysis allowed us to identify how maternal clinical, lifestyle.. contribute to interindividual variability..” but I would say that more than “how” we can talk about “which” determinants contribute. L 434 „Previous studies have focused on measuring free SH in plasma or serum using immunoassays (16).” – when you refer to previous studies you should not use only one citation and, moreover, that opinion is not fully correct since there are studies using LC-MS/MS of GC-MS/MS and you even cite them.
- Some paragraphs are based on only one cited reference which is not enough for proper discussion. Some sentences are more suitable for introduction (e.g. L 464).
- Genetics associations are very poorly discussed - how it may affect whole steroidogenesis and what findings were brought by other studies. Also note - L 568 Kallak et al. (2017) found that rs700518 (CYP19A1) was associated with testosterone (12) – Why is that important? How is it relevant to this study? Was that association positive negative, in what kind of participants?
- Information regarding the approximately 80% of SH variability should be included in the conclusion as a key finding. Currently, the conclusion appears to list all analysed determinants. Although these determinants certainly contribute to SH variability, I believe this is not the central message of your study, which aimed to identify the key environmental, genetic, psychological factors etc.
- I appreciate the description of the novel steroid biomarkers.
Minor comments:
- It should be sex steroids not sexual
- In the Results, many times is written “association” but it is not depicted what kind of association – positive, negative, “good, bad”
- In the Results section Title 2.6. External Replication Analysis (INMA-Sabadell Cohort) refers to INMA cohort but BiSC results are mentioned as well
- Limitation section contains also some generally known facts and not limitations
- L 714 – the sentence is not finished
- L 736 – what does it mean that DNA was extracted randomly?
Author Response
Reviewer Comments
Reviewer 1
Dear Colleagues,
Thank you very much for the interesting study you have submitted for publication. There is a lot of data presented here; however, I have some observations and suggestions that I describe below:
Major comments:
- The main aim and the take home message are not precisely characterised. The abstract lacks this information and introduction contains a bit vague description since it is unusual to set metabolite characterization and analysis as a main goal of the study even when you then continue with information about identification of determinants – I would suggest to rewrite it.
Response: We thank the reviewer for the suggestion. We have revised the abstract and introduction to clearly state the objective of our study, emphasizing that our goal was to comprehensively characterize the maternal steroid hormone metabolome during late pregnancy and identify clinical, lifestyle, and sociodemographic determinants influencing SH metabolism, highlighting the novelty and biological relevance of this approach.
- Abstract is mostly focused on methods but more results and conclusion would be beneficial – in this form it seems like no novelty was brought.
Response: We thank the reviewer for the suggestion. The abstract has been adjusted to reduce methodological details and include more results and conclusions. Key findings, including replicated associations across the two independent Spanish cohorts and the main determinants of SH metabolism, are highlighted. Due to the 200-word limit, detailed effect sizes could not be included, but the main results and their implications are now included.
- I would recommend the introduction to be shortened – it contains a lot of redundant information and should be more addressed. It is questionable if the Table 1 is necessary and it would be beneficial if the information contained would be integrated into the text. Moreover, some facts e.g., vitamin A intake, Black women ethnicity do not serve for the further analysis.
Response: We thank the reviewer for the suggestion and for the opportunity to clarify this point. The introduction has been shortened to remove redundant information and improve focus. Table 1 has been moved to supplementary tables (Table S13), it provides a concise overview of current evidence on steroid hormones and their determinants during pregnancy, which forms the rationale for the variables included in our analysis. The main points from the table have been summarized within the introduction to facilitate readability and to ensure that the reader can understand its relevance.
- In the Methods section, inclusion and exclusion criteria for both cohorts are missing. Authors do not describe if participants had some pathologies etc. At first, it is not clear which cohort is used for what kind of analysis with which determinants since this information is a part of the Results but definitely it should be included into Methods. I highly recommend to add the majority of the subchapter 2.1. Participant´s characteristic into methods because it brings characteristics (as named in the title) and not results. Moreover, Table 2 is missing some characteristics used further in the study like vitamin supplements, diet, sleep, contraceptive etc.
Response: We thank the reviewer for the suggestion and for the opportunity to clarify this point. The Methods section has been updated to clarify inclusion and exclusion criteria for both cohorts. In addition, it is necessary to mention that Figure 2 visualizes participant selection and the statistical workflow. Additional details on exclusion criteria have been added to improve transparency in the method section. The BiSC cohort is clearly indicated as the primary cohort for the main analysis, and the INMA-Sabadell cohort is designated for replication. The statistical analysis section now explicitly specifies which analyses were performed in each cohort, including ExWAS, Elastic Net selection, and generalized linear models for primary (BiSC) and replication (INMA-Sabadell) analyses.
Regarding participants’ characteristics, information on determinant names was removed and incorporated into method section (4.6.1. Data Preparation).
Finally, we have incorporated more information in Table 1.
- Is the selection of SNPs based only on their relation to steroidogenesis? If so, is there a deeper significance, and are they related to, or candidates for, specific pathologies?
Response: We thank the reviewer for the suggestion and for the opportunity to clarify this point. As we mentioned in method section, we selected SNPs previously associated with steroid hormone enzymatic pathways described in the literature, not necessarily associated with specific pathology in all the cases: CYP3A4 (rs2300701, rs806645), CYP1B1 (rs1056836), HSD3B1 (rs1264459), HSD17B2 (rs1937863), CYP17A1 (rs743572), and CYP19A1 (rs10046, rs28757184, rs700518, rs6493497).
- Kallak TK, Hellgren C, Skalkidou A, Sandelin-Francke L, Ubhayasekhera K, Bergquist J, et al. Maternal and female fetal testosterone levels are associated with maternal age and gestational weight gain. Eur J Endocrinol. 2017 Oct 1;177(4):379–88.
- Mullen J, Gadot Y, Eklund E, Andersson A, J. Schulze J, Ericsson M, et al. Pregnancy greatly affects the steroidal module of the Athlete Biological Passport. Drug Test Anal. 2018;10(7):1070–5.
- Hellgren C, Comasco E, Skalkidou A, Sundström-Poromaa I. Allopregnanolone levels and depressive symptoms during pregnancy in relation to single nucleotide polymorphisms in the allopregnanolone synthesis pathway. Horm Behav. 2017 Aug 1;94:106–13.
- Miodovnik A, Diplas AI, Chen J, Zhu C, Engel SM, Wolff MS. Polymorphisms in the maternal sex steroid pathway are associated with behavior problems in male offspring. Psychiatr Genet. 2012 Jun;22(3):115.
- Regarding results, in the section 2.5 p values and a reference to a table should be added into some sections. E.g. “Pregnant women with PCOS diagnosis before pregnancy had higher androgens levels.”, “Regarding maternal education, women with university level studies had lower progesterone metabolite levels but higher total corticosteroid levels compared with women with primary education”. In general, throughout the Results sections, p values are missing
Response: We thank the reviewer for the suggestion and for the opportunity to clarify this point. Detailed statistical results, including effect sizes, 95% confidence intervals, and p-values, were already provided in Supplementary Tables S10–S12 in the original submission. To enhance clarity, we have now added explicit references to these tables throughout the Results section, allowing readers to easily locate the complete statistical information supporting the summarized findings in the main text. Since we are evaluating several steroid hormones, metabolites, and metabolic indicators, presenting the associations for all determinants by each outcome including effect size and p-value in the main text would lead to overloading and make the Results section difficult to read. Therefore, we opted to summarize key findings in the main text while providing full statistical details in the supplementary tables.
- I am not sure if information from different trimesters from two cohorts are even comparable (L 375).
Response: We thank the reviewer for the suggestion and for the opportunity to clarify this point. We discussed about different timepoint in dietary intake (lines 565-567): “Furthermore, some determinants, such as diet and alcohol intake, were not assessed at equivalent gestational ages across cohorts, potentially introducing bias due to temporal variations in maternal physiology and lifestyle during pregnancy”.
- The entire analysis was performed on an impressive BiSC cohort. However, analysis performed on INMA-Sabadell cohort is redundant and does not offer any additional characteristic, contrarily shows opposite results in some ways. What exactly did you want to prove and demonstrate when you did not use the same analysis (Direct replication)? In my opinion, if one wants to replicate a study and calls it a replication analysis it then needs even bigger n, more factors and some add-ons/new parameters which should be added to whole examination and make it more valuable (more conceptual replication). In this case, it was the exact opposite: a smaller sample size and fewer determinants were used, with questionnaires collected at a different time (a different trimester), and different urine samples were taken. Overall, this gives the impression that these might be leftover data from some study, which are ultimately being used as an addendum to another study.
Response: We thank the reviewer for the suggestion and for the opportunity to clarify this point. The main objective of the replication was not to enlarge the sample size or include additional determinants, but rather to assess whether the observed associations could be reproduced in an independent yet comparable cohort using the same analytical LC-MS/MS approach. As highlighted by Perng and Aslibekyan (2020), in omics studies, replication is best achieved when the confirmatory study sample is independent but similar in population characteristics and uses the same biological samples, analytical platforms, and statistical methods. Likewise, Maitre et al. (2023) emphasized that replication across cohorts is a cornerstone of reliability in exposome and metabolomic research.
The INMA-Sabadell cohort was therefore selected because its design, data collection procedures, and urinary SH characterization align with those of BiSC, allowing us to test the reproducibility of associations under comparable conditions. While the sample size was smaller and some determinants were not available, this cohort provided an opportunity to verify whether key associations were consistent in direction and effect size, supporting the robustness of our findings. We acknowledge and discuss in the manuscript the methodological differences between cohorts (e.g., biological matrix: pool weekly urine samples versus spot urine samples), which may partly explain some heterogeneity in results. Finally, both cohorts were analyzed using the same LC–MS/MS method to quantify 50 steroid hormones and quality control procedures, following best practices for replication in quantitative omics research.
In summary, our approach aligns with current definitions of replication in metabolomic studies, which prioritize methodological consistency and independent validation over larger sample size per se.
References:
- Perng W, Aslibekyan S. Find the needle in the haystack, then find it again: Replication and validation in the ‘omics era. Metabolites. 2020;10(7):286. doi:10.3390/metabo10070286
- Maitre L, Jedynak P, Gallego M, et al. Integrating -omics approaches into population-based studies of endocrine disrupting chemicals: A scoping review. Environ Res. 2023;228:115788. doi:10.1016/j.envres.2023.115788
- The authors describe that approximately 80% of SH variability remains unexplained. There may be several factors contributing to this fact, which should be discussed in detail but are not - some are only mentioned in the limitations section. Key aspects that could be addressed include the cumulative effect of various determinants and the influence of pre-pregnancy life.
Response: We thank the reviewer for the suggestion and for the opportunity to clarify this point. In our main analysis, we included key pre-pregnancy factors such as fertility treatment and PCOS diagnosis, which are known to influence steroid hormone variability during pregnancy. In addition, we provide a detailed discussion of the maternal determinants of steroid hormone variability identified in our study, in the context of previous literature. Determinants not assessed such as statin use, maternal microbiome, and environmental exposures are explicitly acknowledged in the limitations and suggested as directions for future research.
(Lines 548-570)
“Our study faced some limitations. Although we considered a wide range of maternal determinants, including sociodemographic, clinical, lifestyle, and genetic factors, the models explained less than 20% of the variability in maternal SH levels. This indicates that other relevant contributors were not included in our analysis. Several additional determinants could help explain the remaining variability such as environmental exposures (air pollution, endocrine-disrupting chemicals, and ambient temperature [61–65]; medications related with lipid metabolism such as statins [1]; and maternal microbiome [66]. Additional unmeasured genetic variants, including those in the fetal genome, may also contribute to inter-individual variability in SH metabolism.
Circadian rhythms, which regulate SH secretion and enzymatic activity, can introduce intra-day variation in SH levels that single-timepoint measurements do not capture. In the BiSC cohort, this variability was reduced by analyzing weekly pooled urine samples. In contrast, the replication cohort (INMA–Sabadell) relied on a single spot urine sample, which may have increased variability and partly explained cohort-specific differences.
The cross-sectional design, with SH measured at a single time point in the third trimester, limits causal inference and the longitudinal changes in SH across pregnancy. Furthermore, some determinants, such as diet and alcohol intake, were not assessed at equivalent gestational ages across cohorts, potentially introducing bias due to temporal variations in maternal physiology and lifestyle during pregnancy. Missing data in genetic SNP, mental health (stress and depression), fertility treatment, medication, PCOS and some OC were present in the replication cohort. Finally, as both cohorts were based in Spain, the findings may not be fully generalizable to other populations.”
- Despite the authors use of a novel approach for SH detection, the effects of common determinants (age, BMI, etc.) are already intensively discussed. Thus, the added novelty should be more emphasized.
Response: We thank the reviewer for the suggestion. We have emphasized the novelty of our approach in the introduction and discussion section.
Discussion section:
- More studies should be used for discussion of the obtained results. In the L 425 is written “This analysis allowed us to identify how maternal clinical, lifestyle.. contribute to interindividual variability.” but I would say that more than “how” we can talk about “which” determinants contribute.
Response: We thank the reviewer for the comment. We modified this line according to your suggestion (Lines 398-400): “Our analysis identified that maternal clinical, lifestyle, sociodemographic, genetic, and fetal factors contribute to interindividual variability in SH metabolism.”
- L 434 „Previous studies have focused on measuring free SH in plasma or serum using immunoassays (16).” – when you refer to previous studies you should not use only one citation and, moreover, that opinion is not fully correct since there are studies using LC-MS/MS of GC-MS/MS and you even cite them.
Response: We thank the reviewer for the comment. We have incorporated more references, including a systematic review that summarizes 33 studies with different analytical methods to quantify SH. Changes have been made in the revised manuscript (Lines 402–413): “Using a highly sensitive targeted metabolomic LC-MS/MS approach, we identified 50 conjugated urinary SH metabolites across major SH classes, including glucocorticoids, androgens, estrogens, and progestogens. Beyond measuring individual metabolites and total class levels, we also estimated functional indices of SH metabolism, such as phase I enzymatic activities and S/G ratios. Previous studies have largely focused on free or unconjugated SH fractions in plasma or serum, using immunoassays, LC-MS/MS or GC-MS [33,39,40]. However, few studies have considered conjugated metabolites including glucuronides, sulfate or sulfoglucoconjugate forms, which represent major end-products of SH metabolism [32,41]. Our study extends this literate by reporting novel metabolites such as pregnanediol sulfoglucoconjugate isomers (5-pregnendiol-SG), and estriol-sulfoglucoconjugate, as well as some bisulfate SH metabolites, a minor fraction of the urinary SH metabolome.”
- Some paragraphs are based on only one cited reference which is not enough for proper discussion. Some sentences are more suitable for introduction (e.g. L 464).
Response. We thank the reviewer for the suggestion and for the opportunity to clarify this point. Due to the novelty of our study and the limited information available on several SH conjugates metabolites during pregnancy, we found limited data in the literature for direct comparison in some determinants. Therefore, in some cases, we explicitly indicated when information in pregnant women was not available and justified our interpretation using data from non-pregnant populations.
In addition, we have deleted or modified certain sentences that were more appropriate for the Introduction section to maintain the focus of the Discussion.
- Genetics associations are very poorly discussed - how it may affect whole steroidogenesis and what findings were brought by other studies. Also note - L 568 Kallak et al. (2017) found that rs700518 (CYP19A1) was associated with testosterone (12) – Why is that important? How is it relevant to this study? Was that association positive or negative, in what kind of participants?
Response. We thank the reviewer for this valuable comment. We have now extended the discussion on the genetic associations and their potential effects on steroidogenesis.
Changes have been made in the revised manuscript (Lines 506–524):
“3.5. Genetic: SNPs from Steroidogenesis Enzymes
Common genetic variants in SRD5A2, CYP17A1, and CYP19A1 may influence multiple enzymatic steps in steroidogenesis, contributing to interindividual variation in maternal steroid profiles during pregnancy. Although genetic data were not included in the replication analysis due to the high percentage of missing values, some maternal SNPs involved in key steroidogenic enzymes showed associations with specific SH and activity ratios in the BiSC cohort.
Variants in SRD5A2 (rs2300701, rs806645) were linked to higher androgen and corticosteroid metabolites, and 5α-reductase activity. SRD5a2 is a critical enzyme in the metabolism of androgens [59]. The CYP17A1 rs743572 variant was associated with both androgens and corticosteroids, suggesting modulation of the 17α-hydroxylase and 17,20-lyase. Within CYP19A1, rs6493497 and rs700518 were associated with higher progestogen metabolites and lower 11β-HSD2 activity. While the mechanistic link is not fully understood, these results highlight that variation in aromatase, involved in androgen-to-estrogen conversion, may be part of interindividual differences in SH metabolism during pregnancy. Kallak et al. (2017) also examined CYP19A1 rs700518 in pregnant women at 35–39 gestational weeks and found that mothers carrying male fetuses with the CC genotype had higher testosterone levels than carriers of the T allele, consistent with reduced aromatase activity [13].”
- Information regarding the approximately 80% of SH variability should be included in the conclusion as a key finding. Currently, the conclusion appears to list all analysed determinants. Although these determinants certainly contribute to SH variability, I believe this is not the central message of your study, which aimed to identify the key environmental, genetic, psychological factors etc.
Response. We thank the reviewer for the suggestion and for the opportunity to clarify this point. We thank the reviewer for this insightful suggestion. We agree that highlighting the unexplained variability in SH is an important point. However, as our analyses were limited to the variables included in the study, we cannot draw conclusions regarding factors that were not directly assessed. We have now revised the conclusion to clarify that the variables evaluated in our study explained less than 20% of the variability in SH levels. We also acknowledge that the remaining ~80% may be influenced by other unmeasured factors that were described in the discussion section. Changes have been made in the revised manuscript.
- I appreciate the description of the novel steroid biomarkers.
Response. We thank the reviewer for your comment.
Minor comments:
- It should be sex steroids not sexual
Response: We thank the reviewer for the comment. We corrected it.
- In the Results, many times is written “association” but it is not depicted what kind of association – positive, negative, “good, bad”.
Response. We thank the reviewer for the suggestion and for the opportunity to clarify this point. In the revised manuscript, we have clarified the directionality of the reported associations. We described whether each determinant was associated with higher or lower levels of SH metabolites or metabolic indicators. At this stage, we cannot characterize these associations as “good” or “bad,” because the outcomes were analyzed as continuous quantitative variables, without reference to specific thresholds defining normal or altered SH levels.
- In the Results section Title 2.6. External Replication Analysis (INMA-Sabadell Cohort) refers to INMA cohort but BiSC results are mentioned as well
Response. We thank the reviewer for the suggestion. We thank the reviewer for pointing this out. To avoid confusion, we have revised Section 2.6 to include only the results from the INMA-Sabadell cohort.
- Limitation section contains also some generally known facts and not limitations
Response. We thank the reviewer for the suggestion. We have revised the Limitations section to focus exclusively on study-specific limitations and removed phrases that reflect generally known facts.
- L 714 – the sentence is not finished
Response. We thank the reviewer for the suggestion. This phrase was removed from the text.
- L 736 – what does it mean that DNA was extracted randomly?
Response. We thank the reviewer for the suggestion and for the opportunity to clarify this point. The sentence has been clarified as follows: “Peripheral blood was collected from mothers in EDTA tubes during pregnancy (12 or 32 weeks) or at delivery, and maternal DNA was extracted from a randomly selected subset of samples for analysis.” (Lines 674-677).

Reviewer 2 Report
Comments and Suggestions for Authors
This manuscript provides one of the most comprehensive analyses to date of the maternal urinary steroid hormone metabolome during late pregnancy. By combining targeted UHPLC-MS/MS quantification of 50 conjugated steroid metabolites with advanced statistical modeling (ExWAS, Elastic Net regression), and replicating key findings in an independent cohort, the authors address an important knowledge gap on the determinants of steroid hormone variability in pregnancy. The study is highly relevant for IJMS and represents a valuable contribution to the field of maternal–fetal endocrinology and exposomics.
Strengths
-
Large and well-characterized cohorts: over 1,200 pregnant women from two independent cohorts, enhancing statistical power and external validity.
-
Advanced analytical approach: comprehensive quantification of conjugated metabolites (sulfates, glucuronides) and calculation of enzymatic activity indices.
-
Robust statistical framework: use of ExWAS with multiple-testing correction and penalized regression (Elastic Net) to identify key determinants.
-
Replication analysis: confirmation of associations in an independent cohort strengthens confidence in the findings.
-
Public health relevance: identification of modifiable factors (BMI, physical activity, smoking) with potential implications for maternal and fetal health.
-
Clear and well-organized writing: introduction and discussion provide thorough context and situate the findings within the existing literature.
Weaknesses / Limitations
-
Limited generalizability: both cohorts are from Spain, which may not capture variability in other ethnic, dietary, or environmental settings.
-
Missing data in replication cohort: several key variables (e.g., genetic polymorphisms, mental health indicators, some obstetric complications) were unavailable, limiting the strength of replication for those determinants.
-
Cross-sectional design: measurements at a single time point (third trimester) do not allow causal inference or assessment of longitudinal trends.
-
Complexity of results: although comprehensive, the large number of associations and detailed tables may be challenging for readers to interpret.
-
Discussion length: the discussion section is somewhat extensive and could be more concise, focusing on the most clinically relevant findings.
Suggestions for Improvement
- Condense the discussion to emphasize the main biological and clinical implications, reducing repetition and excessive detail.
- Clarify the limitations more explicitly, particularly regarding generalizability and the cross-sectional design.
- Improve visualization of results (e.g., simplified heatmaps or summary plots) to enhance accessibility for non-specialist readers.
- Highlight future directions, such as the need for longitudinal studies or investigations in more diverse populations.
- Ensure clarity and accessibility of supplementary material, as the reproducibility of this work relies heavily on detailed tables and figures.
Author Response
Reviewer 2:
This manuscript provides one of the most comprehensive analyses to date of the maternal urinary steroid hormone metabolome during late pregnancy. By combining targeted UHPLC-MS/MS quantification of 50 conjugated steroid metabolites with advanced statistical modeling (ExWAS, Elastic Net regression), and replicating key findings in an independent cohort, the authors address an important knowledge gap on the determinants of steroid hormone variability in pregnancy. The study is highly relevant for IJMS and represents a valuable contribution to the field of maternal–fetal endocrinology and exposomics.
Strengths
- Large and well-characterized cohorts: over 1,200 pregnant women from two independent cohorts, enhancing statistical power and external validity.
- Advanced analytical approach: comprehensive quantification of conjugated metabolites (sulfates, glucuronides) and calculation of enzymatic activity indices.
- Robust statistical framework: use of ExWAS with multiple-testing correction and penalized regression (Elastic Net) to identify key determinants.
- Replication analysis: confirmation of associations in an independent cohort strengthens confidence in the findings.
- Public health relevance: identification of modifiable factors (BMI, physical activity, smoking) with potential implications for maternal and fetal health.
- Clear and well-organized writing: introduction and discussion provide thorough context and situate the findings within the existing literature.
Weaknesses / Limitations
- Limited generalizability: both cohorts are from Spain, which may not capture variability in other ethnic, dietary, or environmental settings.
- Missing data in replication cohort: several key variables (e.g., genetic polymorphisms, mental health indicators, some obstetric complications) were unavailable, limiting the strength of replication for those determinants.
- Cross-sectional design: measurements at a single time point (third trimester) do not allow causal inference or assessment of longitudinal trends.
- Complexity of results: although comprehensive, the large number of associations and detailed tables may be challenging for readers to interpret.
Discussion length: the discussion section is somewhat extensive and could be more concise, focusing on the most clinically relevant findings.
Response: We appreciate the overall consideration given to our manuscript by the reviewer and the opportunity to revise the manuscript.
Suggestions for Improvement
- Condense the discussion to emphasize the main biological and clinical implications, reducing repetition and excessive detail.
Response. We thank the reviewer for the suggestion. The Discussion section has been revised to emphasize the main biological and clinical implications of our findings, reducing repetition and excessive detail.
- Clarify the limitations more explicitly, particularly regarding generalizability and the cross-sectional design.
Response. We thank the reviewer for the suggestion. We have added this suggestion in limitation of the study (Lines 563-570): “The cross-sectional design, with SH measured at a single time point in the third trimester, limits causal inference and the longitudinal changes in SH across pregnancy. Furthermore, some determinants, such as diet and alcohol intake, were not assessed at equivalent gestational ages across cohorts, potentially introducing bias due to temporal variations in maternal physiology and lifestyle during pregnancy. Missing data in genetic SNP, mental health (stress and depression), fertility treatment, medication, PCOS and some OC were present in the replication cohort. Finally, as both cohorts were based in Spain, the findings may not be fully generalizable to other populations.”
- Improve visualization of results (e.g., simplified heatmaps or summary plots) to enhance accessibility for non-specialist readers.
Response. We thank the reviewer for the suggestion. We have incorporated Figure 7, a heatmap to summarize the linear regression models from both cohorts.
- Highlight future directions, such as the need for longitudinal studies or investigations in more diverse populations.
Response. We thank the reviewer for the suggestion. We have added this suggestion to our study (Lines 563-575): “The cross-sectional design, with SH measured at a single time point in the third trimester, limits causal inference and the longitudinal changes in SH across pregnancy. Furthermore, some determinants, such as diet and alcohol intake, were not assessed at equivalent gestational ages across cohorts, potentially introducing bias due to temporal variations in maternal physiology and lifestyle during pregnancy. Missing data in genetic SNP, mental health (stress and depression), fertility treatment, medication, PCOS and some OC were present in the replication cohort. Finally, as both cohorts were based in Spain, the findings may not be fully generalizable to other populations.”
- Ensure clarity and accessibility of supplementary material, as the reproducibility of this work relies heavily on detailed tables and figures.
Response. We thank the reviewer for the suggestion and for the opportunity to clarify this point. We have provided detailed supplementary materials, including full lists of SH metabolites, selection of determinants, and details of data processing. Statistical analysis results from ExWAS, elastic net regression (ENET) with cross-validation and bootstrap results, and LM analyses are included. Due to the large volume of information, especially from ExWAS and ENET, all supplementary tables have been compiled into an Excel file, with the first sheet providing the title list of all tables. Additionally, the scripts for the main analyses (ExWAS, ENET, and linear regression models) have been made publicly available on GitHub: https://github.com/Elaveriano/pregnancy-steroid-determinants
